# EFFICIENT, STABLE, AND ANALYTIC DIFFERENTIATION OF THE SINKHORN LOSS

## ABSTRACT

Optimal transport and the Wasserstein distance have become indispensable building blocks of modern deep generative models, but their computational costs greatly prohibit their applications in statistical machine learning models. Recently, the Sinkhorn loss, as an approximation to the Wasserstein distance, has gained massive popularity, and much work has been done for its theoretical properties. To embed the Sinkhorn loss into gradient-based learning frameworks, efficient algorithms for both the forward and backward passes of the Sinkhorn loss are required. In this article, we first demonstrate issues of the widely-used Sinkhorn's algorithm, and show that the L-BFGS algorithm is a potentially better candidate for the forward pass. Then we derive an analytic form of the derivative of the Sinkhorn loss with respect to the input cost matrix, which results in an efficient backward algorithm. We rigorously analyze the convergence and stability properties of the advocated algorithms, and use various numerical experiments to validate the performance of the proposed methods.

## 1 INTRODUCTION

Optimal transport (OT, Villani, 2009) is a powerful tool to characterize the transformation of probability distributions, and has become an indispensable building block of generative modeling. At the core of OT is the Wasserstein distance, which measures the difference between two distributions. For example, the Wasserstein generative adversarial network (WGAN, Arjovsky et al., 2017) uses the 1-Wasserstein distance as the loss function to minimize the difference between the data distribution and the model distribution, and a huge number of related works emerge afterwards.

Despite the various appealing theoretical properties, one major barrier for the wide applications of OT is the difficulty in computing the Wasserstein distance. For two discrete distributions, OT solves a linear programming problem of $nm$ variables, where $n$ and $m$ are the number of Diracs that define the two distributions. Assuming $n = m$, standard linear programming solvers for OT have a complexity of $\mathcal{O}(n^3 \log n)$ (Pele & Werman, 2009), which quickly becomes formidable as $n$ gets large, except for some special cases (Peyré et al., 2019).

To resolve this issue, many approximate solutions to OT have been proposed, among which the Sinkhorn loss has gained massive popularity (Cuturi, 2013). The Sinkhorn loss can be viewed as an entropic-regularized Wasserstein distance, which adds a smooth penalty term to the original objective function of OT. The Sinkhorn loss is attractive as its optimization problem can be efficiently solved, at least in exact arithmetics, via Sinkhorn's algorithm (Sinkhorn, 1964; Sinkhorn & Knopp, 1967), which merely involves matrix-vector multiplications and some minor operations. Therefore, it is especially suited to modern computing hardware such as the graphics processing units (GPUs). Recent theoretical results show that Sinkhorn's algorithm has a computational complexity of $\mathcal{O}(n^2 \varepsilon^{-2})$ to output an $\varepsilon$-approximation to the unregularized OT (Dvurechensky et al., 2018).

Many existing works on the Sinkhorn loss focus on its theoretical properties, for example Mena & Niles-Weed (2019) and Genevay et al. (2019). In this article, we are mostly concerned with the computational aspect. Since modern deep generative models mostly rely on the gradient-based learning framework, it is crucial to use the Sinkhorn loss with differentiation support. One simple and natural method to enable Sinkhorn loss in back-propagation is to unroll Sinkhorn's algorithm, adding every iteration to the auto-differentiation computing graph (Genevay et al., 2018; Cuturi et al., 2019). However, this approach is typically costly when the number of iterations are large.

Instead, in this article we have derived an *analytic* expression for the derivative of Sinkhorn loss based on quantities computed from the forward pass, which greatly simplifies the back-propagation of the Sinkhorn loss.

More importantly, one critical pain point of the Sinkhorn loss, though typically ignored in theoretical studies, is that Sinkhorn's algorithm is numerically unstable (Peyré et al., 2019). We show in numerical experiments that even for very simple settings, Sinkhorn's algorithm can quickly lose precision. Various stabilized versions of Sinkhorn's algorithm, though showing better stability, still suffer from slow convergence in these cases. In this article, we have rigorously analyzed the solution to the Sinkhorn optimization problem, and have designed both forward and backward algorithms that are provably efficient and stable. The main contribution of this article is as follows:

- We have derived an analytic expression for the derivative of the Sinkhorn loss, which can be efficiently computed in back-propagation.
- We have rigorously analyzed the advocated forward and backward algorithms for the Sinkhorn loss, and show that they have desirable efficiency and stability properties.
- We have implemented the Sinkhorn loss as an auto-differentiable function in the PyTorch and JAX frameworks, using the analytic derivative obtained in this article.

The code to reproduce the results in this article is available at `https://1drv.ms/u/s!ArsORq8a24WmoFjNQtZYE_BERzDQ`.

## 2 THE (SHARP) SINKHORN LOSS AS APPROXIMATE OT

Throughout this article we focus on discrete OT problems. Denote by $\Delta_n = \{w \in \mathbb{R}^n_+ : w^{\mathrm{T}} \mathbf{1}_n = 1\}$ the $n$-dimensional probability simplex, and let $\mu = \sum_{i=1}^n a_i \delta_{x_i}$ and $\nu = \sum_{j=1}^m b_j \delta_{y_j}$ be two discrete probability measures supported on data points $\{x_i\}_{i=1}^n$ and $\{y_j\}_{j=1}^m$, respectively, where $a = (a_1, \ldots, a_n)^{\mathrm{T}} \in \Delta_n$, $b = (b_1, \ldots, b_m)^{\mathrm{T}} \in \Delta_m$, and $\delta_x$ is the Dirac at position $x$. Define $\Pi(a, b) = \{T \in \mathbb{R}^{n \times m}_+ : T\mathbf{1}_m = a, T^{\mathrm{T}}\mathbf{1}_n = b\}$, and let $M \in \mathbb{R}^{n \times m}$ be a cost matrix with entries $M_{ij}, i = 1, \ldots, n, j = 1, \ldots, m$. Without loss of generality we assume that $n \geq m$, as their roles can be exchanged. Then OT can be characterized by the following optimization problem,

$$W(M, a, b) = \min_{P \in \Pi(a,b)} \langle P, M \rangle, \tag{1}$$

where $\langle A, B \rangle = \mathrm{tr}(A^{\mathrm{T}}B)$. An optimal solution to (1), denoted as $P^*$, is typically called an optimal transport plan, and can be viewed as a joint distribution whose marginals coincide with $\mu$ and $\nu$. The optimal value $W(M, a, b) = \langle P^*, M \rangle$ is then called the Wasserstein distance between $\mu$ and $\nu$ if the cost matrix $M$ satisfies some suitable conditions (Proposition 2.2 of Peyré et al., 2019).

As is introduced in Section 1, solving the optimization problem (1) can be difficult even for moderate $n$ and $m$. One approach to regularizing the optimization problem is to add an entropic penalty term to the objective function, leading to the entropic-regularized OT problem (Cuturi, 2013):

$$\tilde{S}_\lambda(M, a, b) = \min_{T \in \Pi(a,b)} \mathcal{S}_\lambda(T) \coloneqq \min_{T \in \Pi(a,b)} \langle T, M \rangle - \lambda^{-1} h(T), \tag{2}$$

where $h(T) = \sum_{i=1}^n \sum_{j=1}^m T_{ij}(1 - \log T_{ij})$ is the entropy term. The new objective function $\mathcal{S}_\lambda(T)$ is $\lambda^{-1}$-strongly convex on $\Pi(a, b)$, so (2) has a unique global solution, denoted as $T_\lambda^*$. In this article, $T_\lambda^*$ is referred to as the Sinkhorn transport plan. The entropic-regularized Wasserstein distance, also known as the Sinkhorn distance or Sinkhorn loss in the literature (Cuturi, 2013), is then defined as

$$S_\lambda(M, a, b) = \langle T_\lambda^*, M \rangle.$$

To simplify the notation, we omit the subscript $\lambda$ in $T_\lambda^*$ hereafter when no confusion is caused. It is worth noting that in the literature, $S_\lambda$ and $\tilde{S}_\lambda$ are sometimes referred to as the *sharp* and *regularized* Sinkhorn loss, respectively. The following proposition from Luise et al. (2018) suggests that $S_\lambda$ achieves a faster rate at approximating the Wasserstein distance than $\tilde{S}_\lambda$. Due to this reason, in this article we focus on the sharp version, and simply call $S_\lambda$ the Sinkhorn loss for brevity.

**Proposition 1** (Luise et al., 2018). *There exist constants $C_1, C_2 > 0$ such that for any $\lambda > 0$, $|S_\lambda(M, a, b) - W(M, a, b)| \leq C_1 e^{-\lambda}$ and $|\tilde{S}_\lambda(M, a, b) - W(M, a, b)| \leq C_2/\lambda$. The constants $C_1$ and $C_2$ are independent of $\lambda$, and depend on $\mu$ and $\nu$.*

# 3 DIFFERENTIATION OF THE SINKHORN LOSS

To use the Sinkhorn loss in deep neural networks or other machine learning tasks, it is also crucial to obtain the derivative of $S_\lambda(M, a, b)$ with respect to its input parameters. Differentiating the Sinkhorn loss typically involves two stages, the forward and backward passes. In the forward pass, the Sinkhorn loss or the transport plan is computed using some optimization algorithm, and in the backward pass the derivative is computed, using either an analytic expression or the automatic differentiation technique. In this section we analyze both passes in details.

Throughout this article we use the following notations. For $x, y \in \mathbb{R}$, $x \wedge y$ means $\min\{x, y\}$. For a vector $v = (v_1, \ldots, v_k)^{\mathrm{T}}$, let $v^{-1} = (v_1^{-1}, \ldots, v_k^{-1})^{\mathrm{T}}$, $\tilde{v} = (v_1, \ldots, v_{k-1})^{\mathrm{T}}$, and denote by $\mathbf{diag}(v)$ the diagonal matrix formed by $v$. Let $u = (u_1, \ldots, u_l)^{\mathrm{T}}$ be another vector, and denote by $u \oplus v$ the $l \times k$ matrix with entries $(u_i + v_j)$. For a matrix $A = (a_{ij}) = (A_1, \ldots, A_k)$ with column vectors $A_1, \ldots, A_k$, let $\tilde{A} = (A_1, \ldots, A_{k-1})$, and $\mathrm{e}_\lambda[A]$ be the matrix with entries $e^{\lambda a_{ij}}$. The symbol $\odot$ denotes the elementwise multiplication operator between matrices or vectors. $\|\cdot\|$ and $\|\cdot\|_F$ stand for the Euclidean norm for vectors and Frobenius norm for matrices, respectively. Finally, we globally define $\eta \equiv \lambda^{-1}$ for simplicity.

## 3.1 ISSUES OF SINKHORN'S ALGORITHM

In the existing literature, one commonly-used method for the forward pass of the Sinkhorn loss is Sinkhorn's algorithm (Sinkhorn, 1964; Sinkhorn & Knopp, 1967). Unlike the original linear programming problem (1), the solution to the Sinkhorn problem has a special structure. Cuturi (2013) shows that the optimal solution $T^*$ can be expressed as

$$T^* = \mathbf{diag}(u^*) M_e \mathbf{diag}(v^*) \tag{3}$$

for some vectors $u^*$ and $v^*$, where $M_e = \left(e^{-\lambda M_{ij}}\right)$. Sinkhorn's algorithm starts from an initial vector $v^{(0)} \in \mathbb{R}_+^m$, and generates iterates $u^{(k)} \in \mathbb{R}_+^n$ and $v^{(k)} \in \mathbb{R}_+^m$ as follows:

$$u^{(k+1)} \leftarrow a \odot [M_e v^{(k)}]^{-1}, \quad v^{(k+1)} \leftarrow b \odot [M_e^{\mathrm{T}} u^{(k+1)}]^{-1}. \tag{4}$$

It can be proved that $u^{(k)} \to u^*$ and $v^{(k)} \to v^*$, and then the Sinkhorn transport plan $T^*$ can be recovered by (3).

Sinkhorn's algorithm is very efficient, as it only involves matrix-vector multiplication and other inexpensive operations. However, one major issue of Sinkhorn's algorithm is that the entries of $M_e = \left(e^{-\lambda M_{ij}}\right)$ may easily underflow when $\lambda$ is large, making some elements of the vectors $M_e v^{(k)}$ and $M_e^{\mathrm{T}} u^{(k+1)}$ close to zero. As a result, some components of $u^{(k+1)}$ and $v^{(k+1)}$ would overflow. Therefore, Sinkhorn's algorithm in its original form is unstable, and in practice the iterations (4) are typically carried out in the logarithmic scale, which we call the Sinkhorn-log algorithm for simplicity. Besides, there are some other works also attempting to improve the numerical stability of Sinkhorn's algorithm (Schmitzer, 2019; Cuturi et al., 2022).

Despite the advancements of Sinkhorn's algorithm, one critical issue observed in practice is that Sinkhorn-type algorithms may be slow to converge, especially for small regularization parameters. This would severely slow down the computation, and may even give misleading results when the user sets a moderate limit on the total number of iterations. Below we show a motivating example to highlight this issue. Consider a triplet $(M, a, b)$ for the Sinkhorn problem, and fix the regularization parameter $\eta$ to be 0.001, where the detailed setting is provided in Appendix B.1. The true $T^*$ matrix is visualized in Figure 1, along with the solutions given by various widely-used algorithms, including Sinkhorn's algorithm, Sinkhorn-log, the stabilized scaling algorithm (Stabilized, Algorithm 2 of Schmitzer, 2019), and the Greenkhorn algorithm (Altschuler et al., 2017; Lin et al., 2022). The maximum number of iterations is 10000 for Greenkhorn and 1000 for other algorithms.

In Figure 1, it is clear that the plans given by Sinkhorn's algorithm and Greenkhorn are farthest to the true value, and Greenkhorn generates NaN values reflected by the white stripes in the plot. In contrast, the stable algorithms Sinkhorn-log and Stabilized greatly improve them. Sinkhorn's algorithm and Sinkhorn-log are equivalent in exact arithmetics, so their numerical differences highlight the need for numerically stable algorithms. However, Sinkhorn-log and Stabilized still have visible inconsistencies with the truth even after 1000 iterations.

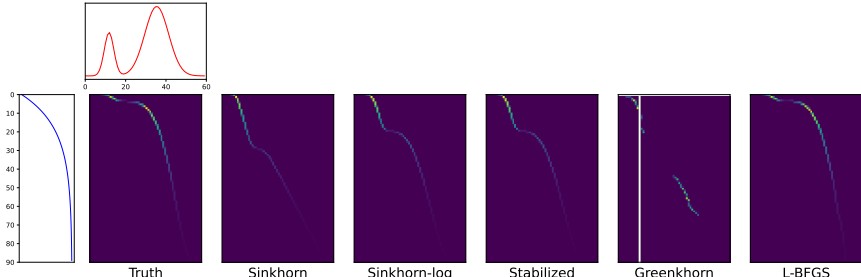

Figure 1: Visualization of Sinkhorn plans computed by different algorithms.

## 3.2 THE ADVOCATED ALTERNATIVE FOR FORWARD PASS

To this end, we advocate an alternative scheme to solve the optimal plan $T^*$, and we show both theoretically and empirically that this method enjoys great efficiency and stability. Consider the dual problem of (2), which has the following form (Proposition 4.4 of Peyré et al., 2019):

$$\max_{\alpha,\beta} \mathcal{L}(\alpha,\beta) := \max_{\alpha,\beta} \alpha^{\mathrm{T}}a + \beta^{\mathrm{T}}b - \eta \sum_{i=1}^{n}\sum_{j=1}^{m} e^{-\lambda(M_{ij}-\alpha_i-\beta_j)}, \quad \alpha \in \mathbb{R}^n, \beta \in \mathbb{R}^m. \quad (5)$$

Let $\alpha^* = (\alpha_1^*, \dots, \alpha_n^*)^{\mathrm{T}}$ and $\beta^* = (\beta_1^*, \dots, \beta_m^*)^{\mathrm{T}}$ be one optimal solution to (5), and then the Sinkhorn transport plan $T^*$ can be recovered as $T^* = \mathrm{e}_\lambda[\alpha^* \oplus \beta^* - M]$. Remarkably, (5) is equivalent to an *unconstrained* convex optimization problem, so a simple gradient ascent method suffices to find its optimal solution. But in practice, quasi-Newton methods such as the limited memory BFGS method (L-BFGS, Liu & Nocedal, 1989) can significantly accelerate the convergence. Using L-BFGS to solve (5) is a known practice (Cuturi & Peyré, 2018; Flamary et al., 2021), but little is known about its stability in solving the regularized OT problem. We first briefly describe the algorithm below, and in Section 4 we rigorously prove that L-BFGS converges fast and generates stable iterates.

It is worth noting we can reduce the number of variables to be optimized in (5) based on the following two findings. First, as pointed out by Cuturi et al. (2019), the variables $(\alpha, \beta)$ have one redundant degree of freedom: if $(\alpha^*, \beta^*)$ is one solution to (5), then so is $(\alpha^* + c\mathbf{1}_n, \beta^* - c\mathbf{1}_m)$ for any $c$. Therefore, we globally set $\beta_m = 0$ without loss of generality. Second, let $\alpha^*(\beta) = \arg\max_\alpha \mathcal{L}(\alpha, \beta)$ for a given $\beta = (\tilde{\beta}^{\mathrm{T}}, \beta_m)^{\mathrm{T}} = (\tilde{\beta}^{\mathrm{T}}, 0)^{\mathrm{T}}$, and define $f(\beta) = -\mathcal{L}(\alpha^*(\beta), \beta)$. Then we only need to minimize $f(\beta)$ with $(m-1)$ free variables to get $\beta^*$, and $\alpha^*$ can be recovered as $\alpha^* = \alpha^*(\beta^*)$. In the appendix we show that $\alpha^*(\beta)$, $f(\beta)$, and $\nabla_{\tilde{\beta}} f$ have simple closed-form expressions:

$$f(\beta) = -\alpha^*(\beta)^{\mathrm{T}}a - \beta^{\mathrm{T}}b + \eta, \qquad \alpha^*(\beta)_i = \eta \log a_i - \eta \log \left[ \sum_{j=1}^{m} e^{\lambda(\beta_j - M_{ij})} \right], \quad (6)$$

$$\nabla_{\tilde{\beta}} f = \tilde{T}(\beta)^{\mathrm{T}} \mathbf{1}_n - \tilde{b}, \qquad T(\beta) = \mathrm{e}_\lambda[\alpha^*(\beta) \oplus \beta - M].$$

With $f(\beta)$ and $\nabla_{\tilde{\beta}} f$, the L-BFGS algorithm can be readily used to minimize $f(\beta)$ and obtain $\beta^*$. Each gradient evaluation requires $\mathcal{O}(mn)$ exponentiation operations, which is comparable to Sinkhorn-log. Although exponentiation is more expensive than matrix-vector multiplication as in Sinkhorn's algorithm, this extra cost can be greatly remedied by modern hardware such as GPUs. On the other hand, we would show in Section 4 that L-BFGS has a strong guarantee on numerical stability, which is critical for many scientific computing problems.

For the motivating example in Section 3.1, we demonstrate the advantage of L-BFGS by showing its transport plan in the rightmost plot of Figure 1. We limit its maximum number of gradient evaluations to 1000, and hence comparable to other methods. Clearly, the L-BFGS solution is visually identical to the ground truth. To study the difference between L-BFGS and Sinkhorn's algorithm in more depth, we compute the objective function value of the dual problem (6) at each iteration for both Sinkhorn-log and L-BFGS. The results are visualized in Figure 2, with three different $\eta$ values, $\eta = 0.1, 0.01, 0.001$. Since each L-BFGS iteration may involve more than one gradient evaluation, for L-BFGS we plot the values against both the outer iteration and gradient evaluation counts.

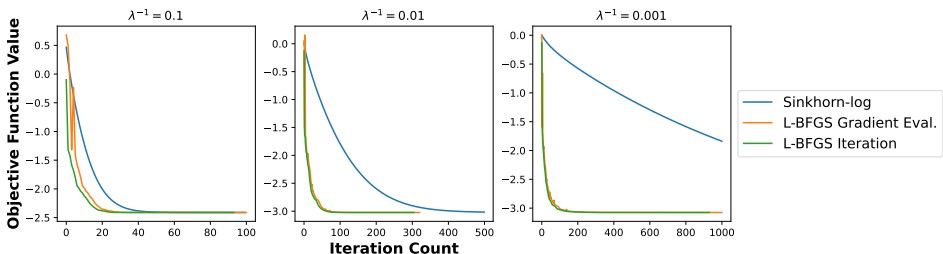

Figure 2: Comparing the convergence speed of Sinkhorn-log and L-BFGS.

Figure 2 gives a clear clue to the issue of Sinkhorn-log: it has a surprisingly slow convergence speed compared to L-BFGS when $\eta$ is small. Theoretically, Sinkhorn algorithms will eventually converge with sufficient iterations, but in practice, a moderate limit on computational budget may prevent them from generating accurate results. To this end, L-BFGS appears to be a better candidate when one needs a small $\eta$ for better approximation to the OT problem. In Appendix B.2 we have designed more experiments to study the forward pass stability and accuracy of different algorithms.

### 3.3 THE ANALYTIC BACKWARD PASS

For the backward pass, one commonly-used method is *unrolled* Sinkhorn's algorithm, which is based on the fact that Sinkhorn's forward pass algorithm is differentiable with respect to $a$, $b$, and $M$. Therefore, one can use automatic differentiation software to compute the corresponding derivatives in the backward pass. This method is used in Genevay et al. (2018) for learning generative models with the Sinkhorn loss, but in practice it may be extremely slow if the forward pass takes a large number of iterations. To avoid the excessive cost of unrolled algorithms, various implicit differentiation methods have been developed for the Sinkhorn loss (Feydy et al., 2019; Campbell et al., 2020; Xie et al., 2020; Eisenberger et al., 2022), but they still do not provide the most straightforward way to compute the gradient.

To this end, we advocate the use of analytic derivatives of the Sinkhorn loss, which solves the optimization problem (2) in the forward pass, and use the optimal dual variables $(\alpha^*, \beta^*)$ for the backward pass. The analytic form for $\nabla_{a,b} S_\lambda(M, a, b)$ has been studied in Luise et al. (2018), and to our best knowledge, few result has been presented for $\nabla_M S_\lambda(M, a, b)$. Our first main theorem, given in Theorem 1, fills this gap and derives the analytic form for $\nabla_M S_\lambda(M, a, b)$.

**Theorem 1.** *For a fixed $\lambda > 0$,*

$$\nabla_M S_\lambda(M, a, b) = T^* + \lambda(s_u \oplus s_v - M) \odot T^*,$$

*where $T^* = \mathrm{e}_\lambda[\alpha^* \oplus \beta^* - M]$, $s_u = a^{-1} \odot (\mu_r - \tilde{T}^* \tilde{s}_v)$, $s_v = (\tilde{s}_v^{\mathrm{T}}, 0)^{\mathrm{T}}$, $\mu_r = (M \odot T^*)\mathbf{1}_m$, $\tilde{\mu}_c = (\tilde{M} \odot \tilde{T}^*)^{\mathrm{T}} \mathbf{1}_n$, $\tilde{s}_v = D^{-1}\left[\tilde{\mu}_c - \tilde{T}^{*\mathrm{T}}(a^{-1} \odot \mu_r)\right]$, and $D = \mathbf{diag}(\tilde{b}) - \tilde{T}^{*\mathrm{T}}\mathbf{diag}(a^{-1})\tilde{T}^*$. In addition, $D$ is positive definite, and hence invertible, for any $\lambda > 0$, $a \in \Delta_n$, $b \in \Delta_m$, and $M$.*

Assuming $n \geq m$, the main computational cost is in forming $D$, which requires $\mathcal{O}(m^2 n)$ operations for matrix-matrix multiplication, and in computing $\tilde{s}_v$, which costs $\mathcal{O}(m^3)$ operations for solving a positive definite linear system.

## 4 CONVERGENCE AND STABILITY ANALYSIS

In Sections 3.2 and 3.3, we have introduced the advocated forward and backward algorithms for the Sinkhorn loss, respectively. In this section, we focus on the theoretical properties of these algorithms, and show that they enjoy provable efficiency and stability. As a first step, we consider the target of the optimization problem (6), and show that $f(\beta)$ has a well-behaved minimizer, which does not underflow or overflow.

**Theorem 2.** *Denote by $f^*$ the minimum value of $f(\beta)$ and $\beta^*$ an optimal solution, and let $\alpha^* = \alpha^*(\beta^*)$. Then $f^* > -\infty$, $\beta^*$ is unique, $\|\alpha^*\| < \infty$, and $\|\beta^*\| < \infty$. In particular, let $I =$*

$\arg\max_i T_{im}^*$, $a_{\max} = \max_i a_i$, and $c = \log(n/b_m)$. Then $\underline{L}_{\alpha_i} \leq L_{\alpha_i} \leq \alpha_i^* \leq U_{\alpha_i}$ and $L_{\beta_j} \leq \beta_j^* \leq U_{\beta_j} \leq \overline{U}_{\beta_j}$ for all $i = 1, \ldots, n$ and $j = 1, \ldots, m$, where

$$U_{\alpha_i} = M_{im} + \eta \cdot \log(a_i \wedge b_m), \qquad\qquad U_{\beta_j} = M_{Ij} - M_{Im} + \eta\left[\log(a_I \wedge b_j) + c\right],$$

$$L_{\alpha_i} = \eta \cdot \log(a_i/m) - \max_j (U_{\beta_j} - M_{ij}), \qquad L_{\beta_j} = \eta \cdot \log(b_j/n) - \max_i (U_{\alpha_i} - M_{ij}),$$

$$\overline{U}_{\beta_j} = \max_i\{M_{ij} - M_{im}\} + \eta\left[\log(a_{\max} \wedge b_j) + c\right], \quad \underline{L}_{\alpha_i} = \eta \cdot \log(a_i/m) - \max_j (\overline{U}_{\beta_j} - M_{ij}).$$

Theorem 2 shows that the optimal dual variables $\gamma^*$ are bounded, and more importantly, the bounds are roughly at the scale of the cost matrix entries $M_{ij}$ and the log-weights $\log(a_i)$ and $\log(b_j)$. Therefore, at least the target of the optimization problem is well-behaved. Moreover, one useful application of Theorem 2 is to impose a box constraint on the variables, adding further stability to the optimization algorithm.

After verifying the properties of the optimal solution, a more interesting and critical problem is to seek a stable algorithm to approach the optimal solution. Indeed, in Theorem 3 we prove that the L-BFGS algorithm for minimizing (6) is one such method.

**Theorem 3.** *Let $\{\tilde{\beta}^{(k)}\}$ be a sequence of iterates generated by the L-BFGS algorithm starting from a fixed initial value $\tilde{\beta}^{(0)}$, and define $\beta^{(k)} = (\tilde{\beta}^{(k)\mathrm{T}}, 0)^{\mathrm{T}}$, $T^{(k)} = \mathrm{e}_\lambda[\alpha^*(\beta^{(k)}) \oplus \beta^{(k)} - M]$, and $f^{(k)} = f(\beta^{(k)})$. Then there exist constants $0 \leq r < 1$ and $C_1, C_2 > 0$ such that for each $k > 0$:*

*(a)* $f^{(k)} - f^* \leq r^k(f^{(0)} - f^*) := \varepsilon^{(k)}$.            *(Linear convergence for the objective value)*

*(b)* $\|\beta^{(k)} - \beta^*\|^2 \leq C_1 \varepsilon^{(k)}$.                 *(Linear convergence for the iterates)*

*(c)* $T^{(k)}\mathbf{1}_m = a$, $\|\nabla_{\tilde{\beta}} f(\beta^{(k)})\|^2 = \|\tilde{T}^{(k)\mathrm{T}}\mathbf{1}_n - \tilde{b}\|^2 \leq C_2 \varepsilon^{(k)}$.    *(Exponential decay of the gradient)*

*(d)* $T_{ij}^{(k)} < \min\{a_i, b_j + \sqrt{C_2 \varepsilon^{(k)}}\}$, $1 \leq j \leq m - 1$.           *($T^{(k)}$ does not overflow)*

*(e)* $\max_j T_{ij}^{(k)} > a_i/m$, $\max_i T_{ij}^{(k)} > (b_j - \sqrt{C_2 \varepsilon^{(k)}})/n$, $1 \leq j \leq m - 1$. *($T^{(k)}$ does not underflow)*

*The explicit expressions for the constants $C_1, C_2, r$ are given in Appendix A.*

Theorem 3 reveals some important information. First, both the objective function value and the iterates have linear convergence speed, so the forward pass using L-BFGS takes $\mathcal{O}(\log(1/\varepsilon))$ iterations to obtain an $\varepsilon$-optimal solution. Second, the marginal error for $\mu$, measured by $\|T^{(k)}\mathbf{1}_m - a\|$, is exactly zero due to the partial optimization on $\alpha$. The other marginal error $\|\tilde{T}^{(k)\mathrm{T}}\mathbf{1}_n - \tilde{b}\|$, which is equal to the gradient norm, is also bounded at *any* iteration, and decays exponentially fast to zero. This validates the numerical stability of the L-BFGS algorithm. Third, the estimated transport plan at any iteration $k$, $T^{(k)}$, is also bounded and stable. This result can be compared with the formulation in (3): it is not hard to find that $u^*$, $v^*$, and $M_e$, when computed individually, can all be unstable due to the exponentiation operations, especially when $\eta$ is small. In contrast, $T^*$ and $T^{(k)}$, thanks to the results in Theorem 3, do not suffer from this issue.

We emphasize that Theorem 3 provides novel results that are not direct consequences of the L-BFGS convergence theorem given in Liu & Nocedal (1989). First, classical theorems only guarantee the convergence of objective function values and iterates as in (a) and (b), whereas we provide richer information such as the marginal errors and transport plans specific to OT problems. More importantly, our results are all nonasymptotic with computable constants. To achieve this, we carefully analyze the eigenvalue structure of the dual Hessian matrix, which is of interest by itself.

Likewise, we show that the derivative of $\nabla_M S_\lambda(M, a, b)$ as in Theorem 1 can also be computed in a numerically stable way. Let $\widehat{\nabla_M S}$ be the $k$-step approximation to $\nabla_M S := \nabla_M S_\lambda(M, a, b)$ using the L-BFGS algorithm, *i.e.*, replacing every $T^*$ in $\nabla_M S$ by $T^{(k)}$. Then we show that the error on gradient also decays exponentially fast.

**Theorem 4.** *Using the symbols defined in Theorems 1 and 3, let $\sigma = 1/\sigma_{\min}(D)$, where $\sigma_{\min}(D)$ is the smallest eigenvalue of $D$. Assume that for some $k_0$,*

$$\varepsilon^{(k_0)} < C_1^{-1}\left[\frac{\min\{1, (6\sigma\|D\|_F)^{-1}\}}{4\lambda}\right]^2,$$

*and then for every $k \geq k_0$, $\|\widehat{\nabla_M S} - \nabla_M S\|_F \leq C_S \sqrt{\varepsilon^{(k)}} = C_S \sqrt{f^{(0)} - f^*} \cdot r^{k/2}$, where the explicit expression for $C_S$ is given in Appendix A. $k_0$ always exists as $\varepsilon^{(k)}$ decays to zero exponentially fast as ensured by Theorem 3(a).*

## 5  APPLICATION: SINKHORN GENERATIVE MODELING

The Sinkhorn loss is useful in unsupervised learning tasks that attempt to match two distributions $p_\theta$ and $p^*$, where $p^*$ stands for the data distribution, and $p_\theta$ is the model distribution. If $p_\theta$ and $p^*$ can be represented or approximated by two discrete measures $\mu_\theta$ and $\nu$, respectively, then one would wish to minimize the Sinkhorn loss $S_\lambda(M_\theta, a_\theta, b_\theta)$ between $\mu_\theta$ and $\nu$, where the cost matrix $M$ and weights $a, b$ may depend on learnable parameters $\theta$. In gradient-based learning frameworks, the key step of seeking the optimal parameter $\theta$ that minimizes $S_\lambda(M_\theta, a_\theta, b_\theta)$ is to compute the gradient $\nabla_\theta S_\lambda(M_\theta, a_\theta, b_\theta)$, which further reduces to evaluating $\nabla_{a,b} S_\lambda(M, a, b)$ and $\nabla_M S_\lambda(M, a, b)$.

Luise et al. (2018) assumes that $M$ is fixed, and studies the gradients $\nabla_{a,b} S_\lambda(M, a, b)$. However, in many generative models it is more important to derive $\nabla_M S_\lambda(M, a, b)$, as the weights $a$ and $b$ are typically fixed, whereas the cost matrix $M$ is computed from the output of some parameterized layers. Consider a data set $X_1, \ldots, X_n$ that follows the distribution $p^*$, $X_i \in \mathbb{R}^p$, and our target is to fit a deep neural network $g_\theta : \mathbb{R}^r \to \mathbb{R}^p$ such that $g_\theta(Z)$ approximately follows $p^*$, where $Z \sim N(0, I_r)$. To this end, we first generate random variates $Z_1, \ldots, Z_m \sim N(0, I_r)$, and let $Y_j = g_\theta(Z_j)$. The two weight vectors are simply taken to be $a = n^{-1} \mathbf{1}_n$, $b = m^{-1} \mathbf{1}_m$, and the cost matrix between $X_i$ and $Y_j$ is given by $M_\theta \in \mathbb{R}^{n \times m}$, where $(M_\theta)_{ij} = \|X_i - g_\theta(Z_j)\|^2$. Then we learn the network parameter $\theta$ by minimizing the Sinkhorn loss $\ell(\theta) = S_\lambda(M_\theta, n^{-1} \mathbf{1}_n, m^{-1} \mathbf{1}_m)$. We show such applications in Section 7.

## 6  RELATED WORK

In Table 1 we list some related work on the differentiation of the Sinkhorn loss, and provide a brief summary of the contribution and limitation of each work. The target of our work is closest to that of Genevay et al. (2018), *i.e.*, to differentiate the Sinkhorn loss with respect to the input cost matrix. However, they use unrolled Sinkhorn's algorithm, so there is no analytic form for $\nabla_M S_\lambda$. Our work is mostly motivated by Luise et al. (2018), but they consider the derivative with respect to the weights instead of the cost matrix. Similarly, Cuturi et al. (2019) and Cuturi et al. (2020) consider gradients on weights and data points, but not the general cost matrix. Campbell et al. (2020); Xie et al. (2020); Eisenberger et al. (2022) all consider the derivative of the transport plan $T^*$ using implicit differentiation. Although this is a more general problem than computing $\nabla_M S_\lambda$, it loses the special structure of the Sinkhorn loss. As a result, the compact matrix form of the derivative presented in Theorem 1 is unique. Furthermore, the storage requirement for our result is $\mathcal{O}(nm)$, whereas some existing works need to store much larger matrices. Finally, very few work has rigorously analyzed the stability of the algorithm, in both the forward and backward passes.

## 7  NUMERICAL EXPERIMENTS

### 7.1  RUNNING TIME OF FORWARD AND BACKWARD PASSES

In our subsequent experiments, we compare three algorithms for differentiating the Sinkhorn loss: the proposed method ("Analytic") using L-BFGS for the forward pass and the analytic derivative for backward pass, the implicit differentiation method ("Implicit"), and the unrolled Sinkhorn algorithm ("Unroll"). Both Implicit and Unroll use Sinkhorn-log for the forward pass, and are implemented in the OTT-JAX library (Cuturi et al., 2022).

We simulate cost matrices of different dimensions, and compare the forward time and total time of different algorithms with both $\eta = 0.1$ and $\eta = 0.01$. The detailed experiment setting and results are given in Appendix B.3, Table 2, and Table 3. The main conclusion is that under the same accuracy level, the forward pass of Analytic is typically faster than those of the other two algorithms, and the backward time is significantly faster. Overall, the proposed analytic differentiation method demonstrates visible advantages on computational efficiency.

Table 1: A brief summary of existing literature on differentiation of Sinkhorn loss.

| Reference | Target | Analytic solution | Derivative w.r.t. the cost matrix | Compact matrix form | Minimum storage requirement | Stability analysis |
|---|---|---|---|---|---|---|
| Genevay et al. (2018) | $\nabla_M S_\lambda$ | ✗ | ✓ | ✗ | ✗ | ✗ |
| Luise et al. (2018) | $\nabla_{a,b} S_\lambda$ | ✓ | ✗ | — | ✓ | ✗ |
| Cuturi et al. (2019) | $\nabla_{a,x} T^*$ | ✗ | ✗ | — | ✗ | ✓ |
| Cuturi et al. (2020) | $\nabla_{b,x} T^*$ | ✓ | ✗ | — | ✓ | ✗ |
| Campbell et al. (2020) | $\nabla_M T^*$ | ✓ | ✓ | ✗ | ✓ | ✗ |
| Xie et al. (2020) | $\nabla_M T^*$ | ✓ | ✓ | ✗ | ✓ | ✗ |
| Eisenberger et al. (2022) | $\nabla_M T^*$ | ✓ | ✓ | ✗ | ✗ | ✗ |
| This article | $\nabla_M S_\lambda$ | ✓ | ✓ | ✓ | ✓ | ✓ |

## 7.2 GENERATIVE MODELS ON TOY DATA SETS

In this section we apply the Sinkhorn loss to generative modeling, and test the accuracy and efficiency of the proposed algorithm. Following the methods in Section 5, we consider a toy data set with $n = 1000$ and $p = 2$ (shown in Figure 3), and we attempt to learn a neural network $g_\theta$ such that $g_\theta$ approximately pushes forward $Z_1, \ldots, Z_m \sim N(0, I_r)$ to the data distribution. In our experiments, $g_\theta$ is a fully-connected ReLU neural network with input dimension $r = 5$ and hidden dimensions 64-128-64. The number of latent data points is $m = 1000$.

In the first setting, we intentionally keep both the observed data and the latent points $Z_1, \ldots, Z_m$ fixed, so that the optimization of $g_\theta$ is a deterministic process without randomness, and the optimization variable obtained from each forward pass is used as the warm start for the next training iteration. At every ten iterations, we compute the 2-Wasserstein distance between the observed data and the pushforward points $\{g_\theta(Z_i)\}$, and we train $g_\theta$ for 2000 iterations using the Adam optimizer with a learning rate of 0.001. This setting, though not common in generative modeling, helps us to monitor the computation of gradients without the impact of random sampling. Moreover, the Wasserstein distance has an achievable lower bound of zero if $g_\theta$ is sufficiently expressive, so we can study the accuracy and efficiency of gradient computation by plotting the metric against running time.

The comparison results for the three algorithms are shown in the second and third plots of Figure 3, from which we have the following findings. First, when $\eta = 0.5$, the 2-Wasserstein distance will increase after certain number of iterations, indicating that the Sinkhorn loss is not an accurate approximation to the Wasserstein distance when $\eta$ is large. Second, it is clear that the Unroll method has the slowest computation, as it needs to unroll a potentially large computational graph in automatic differentiation. Third, the proposed analytic differentiation shows visible advantages on computational efficiency, thanks to the closed-form expression for the derivative.

Finally, to examine the performance of differentiation methods in a genuine generative modeling setting, we use a regular training scheme that randomly samples the observed data and latent points at each iteration. Due to the first finding above, we choose the smaller $\eta$ value for the Sinkhorn loss, and use a mini-batch of size $n = m = 256$ for training. We run 5000 iterations in total, and the metric curves are shown in the last plot of Figure 3. It can be found that the performance of the three algorithms are similar to the fixed-$Z$ case: all three methods properly reduce the Wasserstein distance over time, but the proposed algorithm uses less time to accomplish the computation. Additional experiments on simulated data sets are given in Appendix B.4.

## 7.3 DEEP GENERATIVE MODELS

Finally, we use the Sinkhorn loss to train larger and deeper generative models on the MNIST (Le-Cun et al., 1998) and Fashion-MNIST (Xiao et al., 2017) data sets. In this experiment, we do not pursue training the best possible generative model; instead, we primarily validate our claims on the

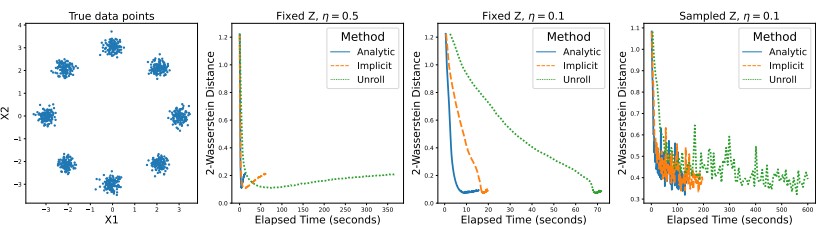

Figure 3: From left to right: the true observed data; the measured Wasserstein distance over time in the fixed $Z$ setting with $\eta = 0.5$; the similar plot for $\eta = 0.1$; the random $Z$ setting with $\eta = 0.1$.

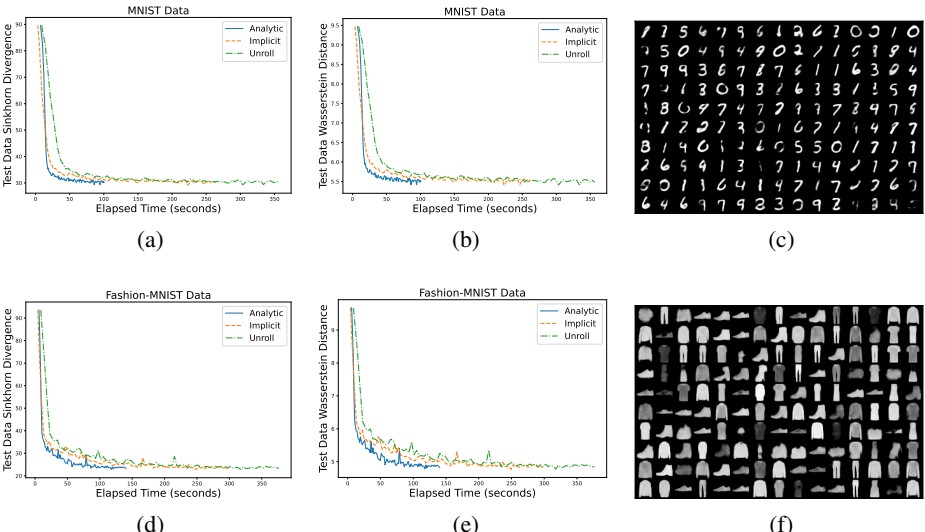

Figure 4: (a)(d) Sinkhorn divergence between randomly sampled test data and model-generated images over iterations. (b)(e) Similar to (a)(d) but using 2-Wasserstein distance. (c)(f) Randomly generated images from the models trained using the analytic differentiation.

efficiency of the proposed analytic differentiation method. The architecture of neural networks and details of the training process are given in Appendix B.5. During the training process, we compute the Sinkhorn divergence (Feydy et al., 2019) and 2-Wasserstein distance between randomly sampled test data and model-generated images over iterations, and Figure 4 show the results of Analytic, Implicit, and Unroll algorithms. The conclusion is similar: the proposed analytic differentiation method achieves the same learning performance as existing methods, but is able to reduce the overall computing time. This once again demonstrates the value of the proposed method.

## 8 CONCLUSION

In this article we study the differentiation of the Sinkhorn loss with respect to its cost matrix, and have derived an analytic form of the derivative, which makes the backward pass of the differentiation easy to implement. Moreover, we study the numerical stability of the forward pass, and rigorously prove that L-BFGS is a stable and efficient algorithm that complements the widely-used Sinkhorn's algorithm and its stabilized versions. In particular, L-BFGS typically converges faster for Sinkhorn problems with a weak regularization. It is worth noting that the proposed analytic differentiation method can be combined with different forward algorithms, and a reasonable scheme is to use L-BFGS for weakly regularized OT problems, and to choose Sinkhorn's algorithm otherwise. The differentiable Sinkhorn loss has many potential applications in generative modeling and permutation learning (Mena et al., 2018), and we anticipate that the technique developed in this article would boost future research on those directions.

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

## A    EXPLICIT EXPRESSIONS FOR CONSTANTS

We first define a few user constants for the L-BFGS algorithm. Let $m_0$ be the maximum number of correction vectors used to construct the BFGS matrix $B_k$, and $c_1 \in (0, 1/2)$, $c_2 \in (c_1, 1)$ are two constants related to the Wolfe condition: we assume the L-BFGS algorithm uses some line search algorithm to select the step sizes $\alpha_k$ that satisfy:

$$f(x_k + \alpha_k d_k) \leq f(x_k) + c_1 \alpha_k g_k^{\mathrm{T}} d_k,$$
$$g(x_k + \alpha_k d_k)^{\mathrm{T}} d_k \geq c_2 g_k^{\mathrm{T}} d_k,$$

where $f(\cdot)$ and $g(\cdot)$ stand for the objective function and gradient function, respectively, $x_k$ is the $k$-th iterate, $g_k = g(x_k)$, $d_k = -B_k^{-1} g_k$ is the search direction, and $B_k$ is the BFGS matrix that approximates the Hessian matrix. $m_0$, $c_1$, and $c_2$ are selected by the user.

For Theorem 3, let $\tilde{\beta}^{(0)}$ be the initial value, and let $\mu = M^{\mathrm{T}} a$ and $u_i = \max_j M_{ij}$, $i = 1, \ldots, n$. Then we define the following constants:

$$U_c = b_m^{-1} \left[ \left( \max_{1 \leq j \leq m-1} \mu_j \right) + \eta \sum_{i=1}^{n} a_i \log a_i - \eta + f(\beta^{(0)}) \right]_+$$

$$A_i = \eta \log a_i - U_c - \eta \log \left( e^{-\lambda(M_{im} + U_c)} + \sum_{j=1}^{m-1} e^{-\lambda M_{ij}} \right), \quad i = 1, \ldots, n$$

$$M_1 = \lambda \cdot \frac{n - m + 2}{2(n - m + 1)} \cdot \min_{1 \leq i \leq n} e^{\lambda(A_i - M_{im})}, \qquad M_2 = \lambda \left[ 1 - \sum_{i=1}^{n} e^{\lambda(A_i - M_{im})} \right]$$

$$M_3 = M_2 - \log M_1 - 1, \qquad M_4 = m - 1 + m_0 M_2 - m_0 \left[ \log M_1 - \log(1 + m_0 M_2) \right]$$

$$C_1 = 2/M_1, \qquad C_2 = 2 M_1^{-1} M_2^2$$

$$r = 1 - c_1(1 - c_2) M_1 / M_2 e^{-(M_3 + M_4)}.$$

For Theorem 4,

$$C_S = 4\lambda \sqrt{C_1} \left[ \|\nabla_M S\|_F + 2\lambda \|T^*\|_F (C_v + C_u) \right]$$
$$C_v = 2\sigma(\|\mu_c\| + 3\|T^{*\mathrm{T}}(a^{-1} \odot \mu_r)\| + 3\|D\|_F \|s_v\|)$$
$$C_u = \|\mu_r\| + 2 C_v \|\mathbf{diag}(a^{-1}) T^*\|_F + \|a^{-1} \odot (T^* s_v)\|.$$

## B    ADDITIONAL EXPERIMENT DETAILS

### B.1    SETTINGS OF THE MOTIVATING EXAMPLE

Consider a small problem of $n = 90$ and $m = 60$. Let $x_i = 5(i - 1)/(n - 1)$, $i = 1, \ldots, n$ be equally-spaced points on $[0, 5]$, and similarly define $y_j = 5(j - 1)/(m - 1)$, $j = 1, \ldots, m$. The cost matrix is set to $M_{ij} = (x_i - y_j)^2$, and the weights $a$ and $b$ are specified as follows. Let $f_1$ be the density function of an exponential distribution with mean 1, and $f_2$ be the density function of a mixture of two normal distributions, $0.2 \cdot N(1, 0.04) + 0.8 \cdot N(3, 0.25)$. And then we set $\tilde{a}_i = f_1(x_i)$, $\tilde{b}_j = f_2(y_j)$, $a_i = \tilde{a}_i / \sum_{k=1}^{n} \tilde{a}_k$, and $b_j = \tilde{b}_j / \sum_{k=1}^{m} \tilde{b}_k$.

We fix the regularization parameter $\eta$ to be 0.001. This value is selected such that the resulting Sinkhorn plan $T_\lambda^*$ is visually close to the OT plan $P^*$. In Figure 5, we show the Wasserstein and Sinkhorn transport plans under different values of $\eta$. It can be seen that when $\eta \leq 0.001$, $T_\lambda^*$ is visually indistinguishable from $P^*$.

We compute the true $T_\lambda^*$ using the $\varepsilon$-scaling algorithm (Algorithm 3 of Schmitzer, 2019). This algorithm is typically accurate, but it requires solving a sequence of Sinkhorn problems with increasing $\lambda$'s, where the solution corresponding to the previous $\lambda$ is used as a warm start for the next one. Therefore, its computational cost is typically large, and it does not compare fairly with other methods. Due to this reason, in this article we mainly use the $\varepsilon$-scaling algorithm to compute high-precision reference values, and do not include it for method comparison.

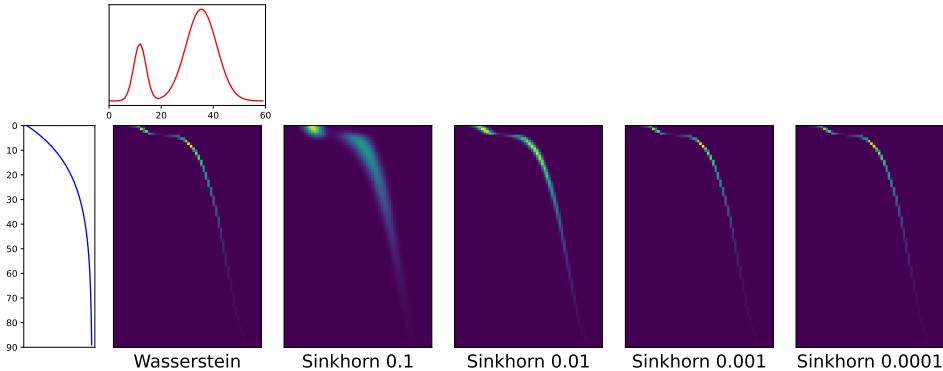

Figure 5: Visualization of Sinkhorn plans with different $\eta$ values.

## B.2 FORWARD PASS STABILITY AND ACCURACY

To further compare the numerical stability and accuracy of different algorithms for computing the Sinkhorn loss, we consider the following experiment. First, we simulate data $X_1, \ldots, X_n \sim f_1(x)$ and $Y_1, \ldots, Y_m \sim f_2(y)$ from some specific distributions $f_1(x)$ and $f_2(y)$, $x, y \in \mathbb{R}^p$, and construct the cost matrix as $M = (M_{ij})$, where $M_{ij} = \|X_i - Y_j\|^2$. The weights are fixed as $a = n^{-1}\mathbf{1}_n$ and $b = m^{-1}\mathbf{1}_m$. For each of the five methods compared in the motivating example in Section 3.1, let $T = (T_{ij})$ be the computed Sinkhorn transport plan and $T^*$ be the true value (computed using the $\varepsilon$-scaling algorithm), and then we compute two types of errors: the error on the transport plan,

$$\text{Err}_{plan}(T) = \sqrt{\sum_{i,j}(T_{ij} - T_{ij}^*)^2},$$

and the error on the Sinkhorn loss value,

$$\text{Err}_{loss}(T) = |\langle T, M \rangle - \langle T^*, M \rangle| = |\langle T - T^*, M \rangle|.$$

For each configuration of the experiment, we simulate the data 100 times, and visualize the distribution of the errors using boxplots.

In our experiment, we fix $n = 150$, $m = 200$, and consider varying dimensions $p = 1, 10, 50$. The Sinkhorn regularization parameters compared are $\eta = 0.01, 0.001$, and for each $\eta$ we set a specific maximum number of iterations for all algorithms. Two data generation models are considered:

(a) Both $f_1(x)$ and $f_2(y)$ are multivariate normal distributions $N(0, I_p)$;

(b) Both $f_1(x)$ and $f_2(y)$ have independent components. Each marginal distribution of $f_1$ is an exponential distribution with mean 1, and each marginal distribution of $f_2$ is a mixture of two normal distributions, $0.2 \cdot N(1, 0.04) + 0.8 \cdot N(3, 0.25)$.

The final results are demonstrated in Figure 6, where all the errors are shown in the log-scale.

In Figure 6, many boxplots for the Stabilized and Greenkhorn algorithms are missing, since they produce all NaN values in the 100 simulations due to numerical overflows. For Sinkhorn, even if it generates no NaNs values explicitly for $\eta = 0.001$, it does not give any meaningful results either. This implies that numerical stability is a critical issue in computing the Sinkhorn loss.

For Sinkhorn-log, it gives reasonably small errors when the regularization is large ($\eta = 0.01$) with sufficient number of iterations, but its accuracy quickly deteriorates when $\eta$ decreases to 0.001. Moreover, we can find that Sinkhorn-log is sensitive to the limits on number of iterations. For example, when $p = 1$ and $\eta = 0.01$, the loss value error can be as small as $10^{-6}$ given 1000 maximum number of iterations, but if we restrict the limit to 200, the error can be as large as $10^{-2}$ or even $10^0$, depending on the data distribution.

In contrast, the difference on maximum number of iterations has a minor effect on the L-BFGS algorithm, indicating that it converges fast, and additional iterations are not needed. These findings demonstrate the advantage of the advocated L-BFGS method in both numerical stability and accuracy.

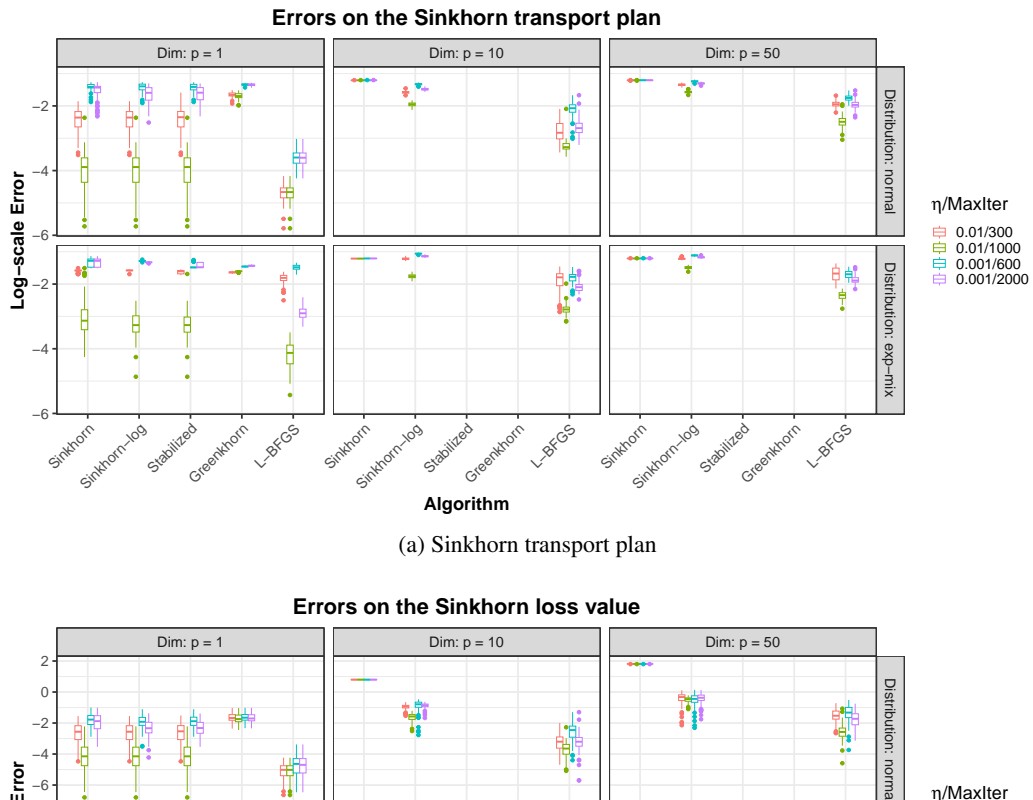

(a) Sinkhorn transport plan

(b) Sinkhorn loss value

Figure 6: Comparing different algorithms on the errors of Sinkhorn transport plan and loss value. The missing boxplots indicate that the corresponding results are NaNs.

## B.3 RUNNING TIME OF FORWARD AND BACKWARD PASSES

In this section we compare the running time of three algorithms on differentiating the Sinkhorn loss.

(a) The proposed algorithm "Analytic": L-BFGS in the forward pass, and analytic differentiation in the backward pass.

(b) "Implicit" implemented in the OTT-JAX library (Cuturi et al., 2022): Sinkhorn-log in the forward pass, and implicit differentiation in the backward pass.

(c) "Unroll" implemented in the OTT-JAX library: Sinkhorn-log in the forward pass, and unrolled automatic differentiation in the backward pass.

We use the second data generation model in Section B.2 to simulate data points, and use the three algorithms above to compute the Sinkhorn loss and its derivative with respect to the cost matrix. For each configuration, we randomly generate the data 100 times, and compute their mean forward time and mean total time. The stopping rule implemented in the OTT-JAX library is $\|T\mathbf{1}_m - a\| +$

$\|T^{\mathrm{T}}\mathbf{1}_n - b\| < \varepsilon_{ott}$, and one of the terms would be exactly zero in the last iteration. The stopping rule for L-BFGS is $\|T^{\mathrm{T}}\mathbf{1}_n - b\|_\infty < \varepsilon_{lbfgs}$. To account for such a difference, we set $\varepsilon_{lbfgs} = 10^{-6}$, and let $\varepsilon_{ott} = \sqrt{\max\{n, m\}} \cdot \varepsilon_{lbfgs}$. In fact, this setting favors the competing method, as its stopping criterion is strictly weaker than the proposed one. To test whether the algorithms actually converge under the given criteria, we also report the number of converging cases within the 100 repetitions.

Results for different data dimensions and regularization parameters are given in Table 2 and Table 3, where the former uses 1000 maximum number of iterations, and the latter uses 10000.

Table 2: Running time of three algorithms for differentiating the Sinkhorn loss, with maximum 1000 iterations.

| | | Mean forward time (ms) | Std. of forward time | Mean total time (ms) | Std. of total time | Converged |
|---|---|---|---|---|---|---|
| $m = n = 64$ $p = 8$ $\eta = 0.1$ | Analytic | 19.07 | 2.11 | **19.33** | 2.13 | 100 |
| | Implicit | 27.77 | 0.69 | 39.66 | 3.41 | 0 |
| | Unroll | 27.53 | 0.13 | 101.30 | 0.36 | 0 |
| $m = n = 64$ $p = 8$ $\eta = 0.01$ | Analytic | 23.21 | 3.45 | **23.48** | 3.46 | 100 |
| | Implicit | 27.33 | 0.05 | 50.52 | 5.19 | 0 |
| | Unroll | 27.39 | 0.12 | 100.66 | 0.39 | 0 |
| $m = n = 128$ $p = 16$ $\eta = 0.1$ | Analytic | 23.79 | 2.96 | **24.19** | 2.96 | 100 |
| | Implicit | 29.81 | 0.61 | 58.21 | 4.91 | 0 |
| | Unroll | 29.75 | 0.06 | 107.18 | 0.19 | 0 |
| $m = n = 128$ $p = 16$ $\eta = 0.01$ | Analytic | 32.24 | 4.99 | **32.68** | 5.00 | 100 |
| | Implicit | 29.73 | 0.05 | 85.56 | 7.48 | 0 |
| | Unroll | 29.65 | 0.08 | 106.66 | 0.46 | 0 |
| $m = n = 256$ $p = 32$ $\eta = 0.1$ | Analytic | 28.60 | 3.82 | **29.40** | 3.85 | 100 |
| | Implicit | 33.69 | 0.32 | 82.53 | 4.21 | 0 |
| | Unroll | 33.60 | 0.23 | 123.25 | 0.07 | 0 |
| $m = n = 256$ $p = 32$ $\eta = 0.01$ | Analytic | 43.94 | 6.08 | **44.86** | 6.12 | 100 |
| | Implicit | 33.75 | 0.20 | 137.46 | 11.66 | 0 |
| | Unroll | 33.78 | 0.10 | 122.99 | 0.27 | 0 |
| $m = n = 512$ $p = 64$ $\eta = 0.1$ | Analytic | 41.35 | 5.64 | **43.05** | 5.53 | 100 |
| | Implicit | 46.06 | 0.05 | 130.96 | 11.97 | 0 |
| | Unroll | 46.02 | 0.04 | 167.64 | 0.22 | 0 |
| $m = n = 512$ $p = 64$ $\eta = 0.01$ | Analytic | 67.76 | 9.65 | **69.42** | 9.65 | 100 |
| | Implicit | 46.08 | 0.05 | 230.34 | 11.72 | 0 |
| | Unroll | 46.09 | 0.05 | 167.68 | 0.20 | 0 |

B.4    MORE EXPERIMENTS ON SIMULATED DATA

We experiment on more simulated data to compare the computational efficiency of Analytic and Implicit. We do not include Unroll since it is too time-consuming. The results are given in Figure 7, as analogues of Figure 3. Three more simulated data sets are studied, and we also include the Sinkhorn loss as a metric to evaluate the model performance over time.

Table 3: Running time of three algorithms for differentiating the Sinkhorn loss, with maximum 10000 iterations.

| | | Mean forward time (ms) | Std. of forward time | Mean total time (ms) | Std. of total time | Converged |
|---|---|---|---|---|---|---|
| $m = n = 64$ $p = 8$ $\eta = 0.1$ | Analytic | 19.01 | 2.10 | **19.27** | 2.10 | 100 |
| | Implicit | 267.95 | 24.57 | 277.86 | 24.58 | 12 |
| | Unroll | 269.49 | 24.76 | 986.46 | 90.67 | 12 |
| $m = n = 64$ $p = 8$ $\eta = 0.01$ | Analytic | 23.28 | 3.46 | **23.54** | 3.48 | 100 |
| | Implicit | 275.42 | 0.34 | 284.84 | 4.74 | 0 |
| | Unroll | 275.54 | 0.49 | 1011.70 | 0.75 | 0 |
| $m = n = 128$ $p = 16$ $\eta = 0.1$ | Analytic | 23.76 | 2.92 | **24.16** | 2.90 | 100 |
| | Implicit | 297.36 | 0.34 | 320.29 | 4.32 | 0 |
| | Unroll | 297.93 | 0.63 | 1074.47 | 1.33 | 0 |
| $m = n = 128$ $p = 16$ $\eta = 0.01$ | Analytic | 32.21 | 5.00 | **32.68** | 5.01 | 100 |
| | Implicit | 297.79 | 0.30 | 324.95 | 9.99 | 0 |
| | Unroll | 297.90 | 0.32 | 1073.24 | 0.84 | 0 |
| $m = n = 256$ $p = 32$ $\eta = 0.1$ | Analytic | 28.72 | 3.84 | **29.57** | 3.83 | 100 |
| | Implicit | 342.23 | 1.53 | 386.80 | 9.22 | 0 |
| | Unroll | 343.18 | 0.70 | 1256.10 | 2.94 | 0 |
| $m = n = 256$ $p = 32$ $\eta = 0.01$ | Analytic | 44.45 | 6.14 | **45.32** | 6.18 | 100 |
| | Implicit | 343.57 | 2.01 | 411.12 | 18.92 | 0 |
| | Unroll | 341.00 | 2.02 | 1253.67 | 3.33 | 0 |
| $m = n = 512$ $p = 64$ $\eta = 0.1$ | Analytic | 41.50 | 5.60 | **43.27** | 5.53 | 100 |
| | Implicit | 469.04 | 0.77 | 552.64 | 15.31 | 0 |
| | Unroll | 465.03 | 1.82 | 1705.51 | 6.36 | 0 |
| $m = n = 512$ $p = 64$ $\eta = 0.01$ | Analytic | 67.97 | 9.69 | **69.68** | 9.67 | 100 |
| | Implicit | 464.59 | 1.04 | 624.28 | 26.48 | 0 |
| | Unroll | 464.16 | 0.55 | 1707.04 | 3.31 | 0 |

## B.5 ARCHITECTURES FOR DEEP GENERATIVE MODELS

For the deep generative models on MNIST and Fashion-MNIST data in Section 7.4, the architectures of the generators are given in Table 4, where $\text{FC}_d$ stands for a fully-connected layer with $d$ output dimensions, and $\text{Conv}_{c,k,s,pin,pout}$ is the transposed convolutional layer with $c$ output channels, kernel size $k$, stride $s$, input padding $pin$, and output padding $pout$.

Both models are trained using the Adam optimizer with learning rate 0.0001, on mini-batches of size 600. The Sinkhorn regularization parameter is set to $\eta = 0.1$, and the training process consists of two stages. In the first stage, we use the squared $L^2$ distance to construct the cost matrix, and in the second stage we switch to the $L^1$ distance. The intuition is that the squared $L^2$ distance has smoother derivatives, thereby making the training more stable in early steps; on the other hand, the $L^1$ distance makes the generated images sharper. The two stages are run for 20 and 30 epochs, respectively. The Wasserstein distances values in Figure 4 are computed in the first stage.

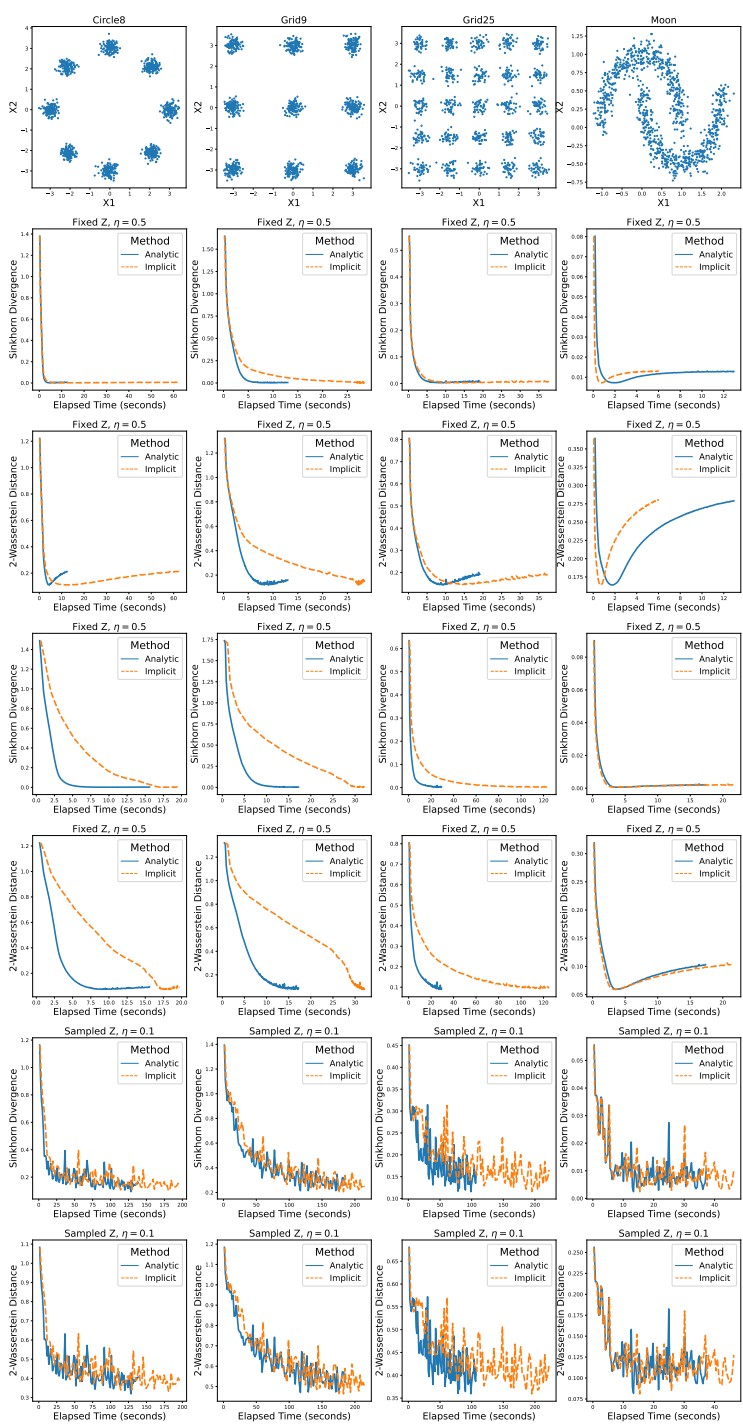

Figure 7: More experiments on simulated data. The plots are analogues of Figure 3, with three additional data sets and the extra Sinkhorn divergence metric to evaluate model performance.

## B.6 APPLYING THE SINKHORN LOSS TO WASSERSTEIN AUTO-ENCODER

Finally, we illustrate the application of the Sinkhorn loss in an Wasserstein auto-encoder (WAE, Tolstikhin et al., 2018) generative model, based on the CelebA data set (Liu et al., 2015). The training data consist of 20000 human face images, each of size $64 \times 64$. The WAE model learns an encoder $Q : \mathbb{R}^p \to \mathbb{R}^r$ and a decoder $G : \mathbb{R}^r \to \mathbb{R}^p$ as a solution to the following optimization

Table 4: Architectures of the neural networks for MNIST and Fashion-MNIST data.

| MNIST | | Fashion-MNIST | |
|---|---|---|---|
| Layer | Output shape | Layer | Output shape |
| Input $z$ | 16 | Input $z$ | 32 |
| $\text{FC}_{3136} + \text{ReLU}$ | 3136 | $\text{FC}_{6272} + \text{ReLU}$ | 6272 |
| Reshape | $64 \times 7 \times 7$ | Reshape | $128 \times 7 \times 7$ |
| $\text{Conv}_{32,4,2,1,0} + \text{ReLU}$ | $32 \times 14 \times 14$ | $\text{Conv}_{64,4,2,1,0} + \text{ReLU}$ | $64 \times 14 \times 14$ |
| $\text{Conv}_{32,4,2,1,1} + \text{ReLU}$ | $32 \times 29 \times 29$ | $\text{Conv}_{64,4,2,1,1} + \text{ReLU}$ | $64 \times 29 \times 29$ |
| $\text{Conv}_{1,4,1,2,0}$ | $1 \times 28 \times 28$ | $\text{Conv}_{1,4,1,2,0}$ | $1 \times 28 \times 28$ |

problem:

$$\min_{Q,G} \mathbb{E}_{p_X} \|X - G(Q(X))\|^2 + \xi \cdot \mathcal{D}(p_Z, p_{Q(X)}), \tag{7}$$

where $p_X$, $p_{Q(X)}$, and $p_Z$ are the distribution of data points $X$, the distribution of $Q(X)$, and a pre-specified latent distribution, respectively, and $\xi$ is a regularization parameter to balance the two terms. The first term in (7) is the reconstruction error, and the second term quantifies the divergence of the distribution of $Q(X)$ to the latent distribution $p_Z$. We simply take $p_Z$ to be a multivariate standard normal distribution, and use the Sinkhorn divergence (Genevay et al., 2018) to define $\mathcal{D}(\cdot)$:

$$\mathcal{D}(\mu, \nu) = S_\lambda(\mu, \nu) - \frac{1}{2} S_\lambda(\mu, \mu) - \frac{1}{2} S_\lambda(\nu, \nu),$$

where $\mu$ and $\nu$ are two discrete distributions, and with slight abuse of notation, $S_\lambda(\mu, \nu)$ is the Sinkhorn loss studied in this article. In actual implementation, $\mu$ and $\nu$ are Diracs of data points from $p_Z$ and $p_{Q(X)}$, respectively. To generate new images, we sample latent data points $Z_1, \ldots, Z_n$ from the latent distribution $p_Z$, and then pass them to the generator to obtain images $Y_i = G(Z_i)$, $i = 1, \ldots, n$.

Since the focus of this article is on the computation and differentiation of the Sinkhorn loss, we do not attempt to build and train the model with full complexity. Instead, to compare computing time, we only run 10 epochs for illustration purpose. The architectures of the encoder and the decoder are given in Table 5, and we set the latent dimension to $r = 64$, the WAE regularization parameter to $\xi = 1$, and the Sinkhorn loss parameter to $\eta = 0.1$. We use the Adam optimizer with a learning rate of 0.001 and a mini-batch size of 500. We have found that the Implicit method generates NaNs after 72 mini-batch iterations, so we only show the results for the proposed Analytic algorithm and the existing Unroll method. Figure 8(a) shows the training process of the WAE model based on the Sinkhorn divergence for 10 epochs, and Figure 8(b) shows the randomly generated images using the Analytic method after 50 epochs. It is clear that the proposed Analytic method is more efficient than the Unroll method in training.

Table 5: Architectures of the neural networks for CelebA data.

| Encoder | | Decoder | |
|---|---|---|---|
| Layer | Output shape | Layer | Output shape |
| Input $x$ | $3 \times 64 \times 64$ | Input $z$ | 64 |
| $\text{Conv}_{32,4,2,1,0} + \text{ReLU}$ | $32 \times 32 \times 32$ | $\text{Conv}_{256,4,1,0,0} + \text{ReLU}$ | $256 \times 4 \times 4$ |
| $\text{Conv}_{64,4,2,1,0} + \text{ReLU}$ | $64 \times 16 \times 16$ | $\text{Conv}_{128,4,2,1,0} + \text{ReLU}$ | $128 \times 8 \times 8$ |
| $\text{Conv}_{128,4,2,1,0} + \text{ReLU}$ | $128 \times 8 \times 8$ | $\text{Conv}_{64,4,2,1,0} + \text{ReLU}$ | $64 \times 16 \times 16$ |
| $\text{Conv}_{256,4,2,1,0} + \text{ReLU}$ | $256 \times 4 \times 4$ | $\text{Conv}_{32,4,2,1,0} + \text{ReLU}$ | $32 \times 32 \times 32$ |
| $\text{Conv}_{64,4,2,0,0}$ | $64 \times 1 \times 1$ | $\text{Conv}_{3,4,2,1,0}$ | $3 \times 64 \times 64$ |

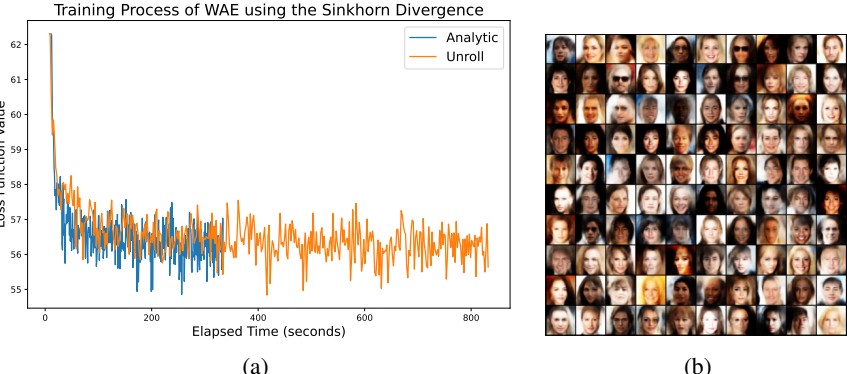

(a)                                                (b)

Figure 8: (a) Training process of WAE for the CelebA data for 10 epochs, based on the Sinkhorn divergence. (b) Randomly generated images using the Analytic method after 50 epochs of training.

## B.7 COMPUTING ENVIRONMENT

All of the experiments in this article were conducted on a personal computer with an Intel i9-11900K CPU and an Nvidia RTX 3090 GPU.

For the motivating example in Section 3.1 and the experiments in Section B.2, results for Sinkhorn's algorithm, Sinkhorn-log, Stabilized, and Greenkhorn are computed using the Python Optimal Transport (POT) library (Flamary et al., 2021). For the experiments in Sections 7.2, 7.3, B.3, B.4, and B.6, the Implicit and Unroll algorithms are implemented in the OTT-JAX library (Cuturi et al., 2022).

## C  PROOFS OF THEOREMS

### C.1  TECHNICAL LEMMAS

In this section we state a few technical lemmas that are used to prove our main theorems. Lemma 1 to Lemma 3 below are standard conclusions in vector calculus, Lemma 4 and Lemma (5) are derived from the eigenvalue theory, and Lemma (7) and Lemma (8) are related to the Sinkhorn problem.

We first introduce the following notations. For any $x \in \mathbb{R}$, let $[x]_+ = \max\{x, 0\}$. For a matrix $A = (a_{ij}) \in \mathbb{R}^{n \times m}$, the vectorization operator, $\mathbf{vec}(A)$, creates a vector by stacking the column vectors of $A$ together, *i.e.*,

$$\mathbf{vec}(A) = (a_{11}, \ldots, a_{n1}, a_{12}, \ldots, a_{n2}, \ldots, a_{1m}, \ldots, a_{nm})^{\mathrm{T}}.$$

For two matrices $A = (a_{ij}) \in \mathbb{R}^{n \times m}$ and $B \in \mathbb{R}^{p \times q}$, the Kronecker product of $A$ and $B$ is defined as

$$A \otimes B = \begin{pmatrix} a_{11}B & \cdots & a_{1m}B \\ \vdots & \ddots & \vdots \\ a_{n1}B & \cdots & a_{nm}B \end{pmatrix}.$$

For a differentiable vector-valued function $f : \mathbb{R}^n \to \mathbb{R}^m$, the partial derivative of $f$ with respect to $x$ is defined as

$$\frac{\partial f(x)}{\partial x^{\mathrm{T}}} = \begin{pmatrix} \frac{\partial f_1(x)}{\partial x_1} & \cdots & \frac{\partial f_1(x)}{\partial x_n} \\ \vdots & \ddots & \vdots \\ \frac{\partial f_m(x)}{\partial x_1} & \cdots & \frac{\partial f_m(x)}{\partial x_n} \end{pmatrix}.$$

We use $I_n$ to denote the $n \times n$ identity matrix, and $\sigma_{\max}(\cdot)$ and $\sigma_{\min}(\cdot)$ stand for the largest and smallest eigenvalues of some symmetric matrix, respectively.

**Lemma 1.** *Given two matrices $A \in \mathbb{R}^{m \times n}$ and $B \in \mathbb{R}^{n \times r}$,*

$$\mathbf{vec}(AB) = (I_r \otimes A)\mathbf{vec}(B) = (B^{\mathrm{T}} \otimes I_m)\mathbf{vec}(A).$$

**Lemma 2.** *Let $f : \mathbb{R}^m \to \mathbb{R}^n$ and $g : \mathbb{R}^m \to \mathbb{R}^n$ be two vector-valued differentiable functions of $x \in \mathbb{R}^m$. Then*

$$\frac{\partial}{\partial x^{\mathrm{T}}} f(x)^{\mathrm{T}} g(x) = g(x)^{\mathrm{T}} \frac{\partial f(x)}{\partial x^{\mathrm{T}}} + f(x)^{\mathrm{T}} \frac{\partial g(x)}{\partial x^{\mathrm{T}}}.$$

**Lemma 3.** *Let $f : \mathbb{R}^m \to \mathbb{R}^l$ and $g : \mathbb{R}^m \to \mathbb{R}^r$ be two vector-valued differentiable functions of $x \in \mathbb{R}^m$. Then*

$$\frac{\partial}{\partial x^{\mathrm{T}}} \mathbf{vec}(f(x)g(x)^{\mathrm{T}}) = (g(x) \otimes I_l) \frac{\partial f(x)}{\partial x^{\mathrm{T}}} + (I_r \otimes f(x)) \frac{\partial g(x)}{\partial x^{\mathrm{T}}}.$$

**Lemma 4.** *Let $A$ and $B$ be two $n \times n$ positive definite matrices, and let $\alpha_1 \geq \cdots \geq \alpha_n > 0$ and $\beta_1 \geq \cdots \geq \beta_n > 0$ be the ordered eigenvalues of $A$ and $B$, respectively. Then for any $x \in \mathbb{R}^n$,*

$$x^{\mathrm{T}} A^{1/2} B A^{1/2} x \leq \alpha_1 \beta_1 \|x\|^2.$$

*Proof.* Let $U_{1 \times n} = x^{\mathrm{T}} A^{1/2}$, and then $u := UU^{\mathrm{T}} = x^{\mathrm{T}} A x \leq \alpha_1 \|x\|^2$, and $UBU^{\mathrm{T}} = x^{\mathrm{T}} A^{1/2} B A^{1/2} x$. By Theorem A.4 in page 788 of Marshall et al. (2011), we immediately get

$$UBU^{\mathrm{T}} = \mathrm{tr}(UBU^{\mathrm{T}}) \leq \beta_1 u \leq \alpha_1 \beta_1 \|x\|^2.$$

$\square$

**Lemma 5.** *Let $A$ and $B$ be two symmetric matrices of the same size. Then*

$$\sigma_{\min}(A) + \sigma_{\min}(B) \leq \sigma_{\min}(A + B) \leq \sigma_{\max}(A + B) \leq \sigma_{\max}(A) + \sigma_{\max}(B).$$

*Proof.* Using the well-known identity $\sigma_{\max}(A) = \max_{\|x\|=1} x^{\mathrm{T}} A x$, we have

$$\sigma_{\max}(A + B) = \max_{\|x\|=1} x^{\mathrm{T}}(A + B)x \leq \max_{\|x\|=1} x^{\mathrm{T}} A x + \max_{\|x\|=1} x^{\mathrm{T}} B x = \sigma_{\max}(A) + \sigma_{\max}(B).$$

Applying the inequality above to $-(A + B)$, we get the result on the opposite direction. $\square$

**Lemma 6.** *Let $x = (x_1, \ldots, x_n)^{\mathrm{T}}$ and $y = (y_1, \ldots, y_n)^{\mathrm{T}}$ be two vectors. Define $\mathrm{LSE}(x) = \log \sum_{i=1}^{n} e^{x_i}$, and then for any $x, y \in \mathbb{R}^n$,*

$$-\|x - y\|_\infty \leq \min_i (x_i - y_i) \leq \mathrm{LSE}(x) - \mathrm{LSE}(y) \leq \max_i (x_i - y_i) \leq \|x - y\|_\infty.$$

*Proof.* It is easy to find that $\nabla_x \mathrm{LSE}(x) = \mathrm{softmax}(x) = (s_1, \ldots, s_n)^{\mathrm{T}}$ is the softmax function, where $s_i = e^{x_i} / \sum_{k=1}^{n} e^{x_k} \in (0, 1)$. By the mean value theorem, we have

$$\mathrm{LSE}(x) - \mathrm{LSE}(y) = \mathrm{softmax}(wx + (1 - w)y)^{\mathrm{T}}(x - y)$$

for some $0 < w < 1$. Let $z = \mathrm{softmax}(wx + (1 - w)y)$, and then

$$\mathrm{LSE}(x) - \mathrm{LSE}(y) = \sum_{i=1}^{n} z_i(x_i - y_i) \leq \left[\max_i (x_i - y_i)\right] \cdot \sum_{i=1}^{n} z_i = \max_i (x_i - y_i) \leq \|x - y\|_\infty,$$

and similarly, $\mathrm{LSE}(x) - \mathrm{LSE}(y) \geq \min_i(x_i - y_i) \geq -\|x - y\|_\infty$. $\square$

**Lemma 7.** *Let $f(\beta)$ be defined as in (6), and let $\mu = M^{\mathrm{T}} a$ and $u_i = \max_j M_{ij}$, $i = 1, \ldots, n$. If $f(\beta) \leq c$, then we have $\max_j \beta_j \leq U_c$ and $\min_j \beta_j \geq L_c$, where*

$$U_c = b_m^{-1} \left[ \left( \max_{1 \leq j \leq m-1} \mu_j \right) + \eta \sum_{i=1}^{n} a_i \log a_i - \eta + c \right]_+ \tag{8}$$

$$L_c = - \left( \min_{1 \leq j \leq m-1} b_j \right)^{-1} \left[ \sum_{i=1}^{n} a_i u_i + \eta \sum_{i=1}^{n} a_i \log a_i - \eta + c \right]_+ . \tag{9}$$

*Proof.* By definition,

$$f(\beta) = \eta \sum_{i=1}^{n} a_i \log \left[ \sum_{j=1}^{m} e^{\lambda(\beta_j - M_{ij})} \right] - \eta \sum_{i=1}^{n} a_i \log a_i - \beta^{\mathrm{T}} b + \eta.$$

If $f(\beta) \leq c$, then

$$c_0 := c + \eta \sum_{i=1}^{n} a_i \log a_i - \eta \geq \eta \sum_{i=1}^{n} a_i \log \left[ \sum_{j=1}^{m} e^{\lambda(\beta_j - M_{ij})} \right] - \beta^{\mathrm{T}} b.$$

By definition we have $\beta_m = 0$, and let $J = \arg\max_{1 \leq j \leq m-1} \beta_j$. Then

$$c_0 \geq \eta \sum_{i=1}^{n} a_i \log \left[ e^{\lambda(\beta_J - M_{iJ})} \right] - \beta^{\mathrm{T}} b = \sum_{i=1}^{n} a_i (\beta_J - M_{iJ}) - \sum_{j=1}^{m-1} \beta_j b_j$$

$$\geq \beta_J - \sum_{i=1}^{n} a_i M_{iJ} - \beta_J \sum_{j=1}^{m-1} b_j = b_m \beta_J - \mu_J \geq b_m \beta_J - \max_{1 \leq j \leq m-1} \mu_j,$$

which verifies (8) by noting that $\max_j \beta_j = [\beta_J]_+$.

Next, let $K = \arg\min_{1 \leq j \leq m-1} \beta_j$. We can assume that $\beta_K < 0$, since otherwise the trivial bound $\min_j \beta_j = \beta_m = 0$ is already met. Consider the sets $S_+ = \{j : \beta_j > 0\}$ and $S_- = \{j : \beta_j < 0\}$. Then clearly,

$$\beta^{\mathrm{T}} b = \beta_K b_K + \sum_{\substack{j \neq K \\ j \in S_+}} \beta_j b_j + \sum_{\substack{j \neq K \\ j \in S_-}} \beta_j b_j \leq \beta_K b_K + [\beta_J]_+ \cdot \sum_{\substack{j \neq K \\ j \in S_+}} b_j + 0 \leq \beta_K b_K + [\beta_J]_+ (1 - b_m).$$

Also note that $\log(\sum_{j=1}^{m} e^{x_j}) \geq \max_j x_j$ for any $x_1, \ldots, x_n \in \mathbb{R}$, so

$$\eta \sum_{i=1}^{n} a_i \log \left[ \sum_{j=1}^{m} e^{\lambda(\beta_j - M_{ij})} \right] \geq \eta \sum_{i=1}^{n} a_i \log \left[ \sum_{j=1}^{m} e^{\lambda(\beta_j - u_i)} \right] = \eta \sum_{i=1}^{n} a_i \left[ \log \left( \sum_{j=1}^{m} e^{\lambda \beta_j} \right) - \lambda u_i \right]$$

$$= \eta \log \left( \sum_{j=1}^{m} e^{\lambda \beta_j} \right) - \sum_{i=1}^{n} a_i u_i \geq \eta \cdot \max_j (\lambda \beta_j) - \sum_{i=1}^{n} a_i u_i$$

$$= \max_j \beta_j - \sum_{i=1}^{n} a_i u_i = [\beta_J]_+ - \sum_{i=1}^{n} a_i u_i.$$

As a result,

$$c_0 \geq \eta \sum_{i=1}^{n} a_i \log \left[ \sum_{j=1}^{m} e^{\lambda(\beta_j - M_{ij})} \right] - \beta^{\mathrm{T}} b \geq [\beta_J]_+ - \sum_{i=1}^{n} a_i u_i - \beta_K b_K - [\beta_J]_+ (1 - b_m)$$

$$= b_m [\beta_J]_+ - \sum_{i=1}^{n} a_i u_i - \beta_K b_K \geq - \sum_{i=1}^{n} a_i u_i - \beta_K b_K,$$

and then

$$\beta_K \geq -b_K^{-1} \left[ \sum_{i=1}^{n} a_i u_i + c_0 \right]_+ \geq - \left( \min_{1 \leq j \leq m-1} b_j \right)^{-1} \left[ \sum_{i=1}^{n} a_i u_i + c_0 \right]_+,$$

which verifies (9). □

**Lemma 8.** *Let $T$ be an $n \times m$ matrix with strictly positive entries, and suppose that $n \geq m$. Define $\mu = T\mathbf{1}_m$, $\nu = T^{\mathrm{T}} \mathbf{1}_n$, and*

$$H = \begin{pmatrix} \mathbf{diag}(\mu) & \tilde{T} \\ \tilde{T}^{\mathrm{T}} & \mathbf{diag}(\tilde{\nu}) \end{pmatrix}, \qquad D = \mathbf{diag}(\tilde{\nu}) - \tilde{T}^{\mathrm{T}} \mathbf{diag}(\mu)^{-1} \tilde{T}.$$

*Then*

$$\sigma_{\max}(D) \leq \max_{1 \leq j \leq m-1} \nu_j,$$

$$\sigma_{\min}(D) \geq \sigma_{\min}(H) \geq \frac{n-m+2}{2(n-m+1)} \cdot \min_{1 \leq i \leq n} T_{im},$$

$$\sigma_{\min}(D) \geq \min_{1 \leq i \leq m-1} \sum_{j=1}^{m-1} D_{ij} = \min_{1 \leq j \leq m-1} \sum_{i=1}^{n} \mu_i^{-1} T_{ij} T_{im}.$$

*Proof.* Consider the matrix $S = H - sJ$, where $J$ is an $(n+m-1) \times (n+m-1)$ matrix filled of ones, and $s$ is a positive scalar. Let

$$R_k = \sum_{j \neq k} |S_{kj}|, \quad k = 1, \ldots, n+m-1.$$

Suppose $s \leq \min_{1 \leq i \leq n, 1 \leq j \leq m-1} T_{ij}$, and then for $k = 1, \ldots, n$, it is easy to find that

$$R_k = (n-1)s + \sum_{j=1}^{m-1}(T_{kj} - s) = (n-1)s + \mu_k - T_{km} - (m-1)s = (n-m)s + \mu_k - T_{km},$$

and for $k = n+1, \ldots, n+m-1$,

$$R_k = (m-2)s + \sum_{i=1}^{n}(T_{i,k-n} - s) = (m-2)s + \nu_{k-n} - ns = (m-n-2)s + v_{k-n}.$$

Then it is easy to see that

$$S_{kk} - R_k = \begin{cases} \mu_k - R_k = T_{km} - (n-m)s, & k = 1, \ldots, n \\ \nu_{k-n} - R_k = (n+2-m)s, & k = n+1, \ldots, n+m-1 \end{cases}.$$

Let

$$\min_{1 \leq i \leq n} T_{im} - (n-m)s = (n+2-m)s,$$

and then $s = \min_{1 \leq i \leq n} T_{im}/(2n - 2m + 2)$, and

$$S_{kk} - R_k \geq L := \frac{n-m+2}{2(n-m+1)} \cdot \min_{1 \leq i \leq n} T_{im} > 0$$

for all $k$. By the Gershgorin circle theorem, every eigenvalue of $S$ must be greater than $L$. Since $H = S + sJ$ and $J$ is nonnegative definite, we also have $\sigma_{\min}(H) \geq L > 0$, implying that $H$ is positive definite.

For the second formula, it is easy to find that the $D$ matrix is the Schur complement of the block $\mathbf{diag}(\mu)$ of the $H$ matrix. So by Theorem 3.1 of Fan (2002), we have $\sigma_{\min}(D) \geq \sigma_{\min}(H)$ and $\sigma_{\max}(D) \leq \sigma_{\max}(\mathbf{diag}(\tilde{\nu})) = \max_{1 \leq j \leq m-1} \nu_j$.

Finally, let $c = \max_{1 \leq j \leq m-1} \nu_j$, and then $D$ can be expressed as $D = cI_{m-1} - B$, where $B = \tilde{T}^{\mathrm{T}} \mathbf{diag}(\mu)^{-1} \tilde{T} + \mathbf{diag}(c\mathbf{1}_{m-1} - \tilde{\nu})$ is a matrix that have nonnegative entries. In addition, we have proved that $D$ is positive definite, so $D$ is a nonsingular $M$-matrix by the definition in Tian & Huang (2010). Then Theorem 3.2 of Tian & Huang (2010) shows that

$$\sigma_{\min}(D) \geq \min_{1 \leq i \leq m-1} \sum_{j=1}^{m-1} D_{ij}.$$

Let $\delta = D\mathbf{1}_{m-1}$, and then clearly $\min_{1 \leq i \leq m-1} \sum_{j=1}^{m-1} D_{ij} = \min_i \delta_i$. Note that

$$\delta = D\mathbf{1}_{m-1} = \tilde{\nu} - \tilde{T}^{\mathrm{T}} \mathbf{diag}(\mu)^{-1} \tilde{T} \mathbf{1}_{m-1} = \tilde{\nu} - \tilde{T}^{\mathrm{T}} \mathbf{diag}(\mu)^{-1}(T\mathbf{1}_m - T_m)$$

$$= \tilde{\nu} - \tilde{T}^{\mathrm{T}} \mathbf{diag}(\mu)^{-1}(\mu - T_m) = \tilde{\nu} - \tilde{T}^{\mathrm{T}} \mathbf{1}_n + \tilde{T}^{\mathrm{T}} \mathbf{diag}(\mu)^{-1} T_m = \tilde{T}^{\mathrm{T}} \mathbf{diag}(\mu)^{-1} T_m,$$

where $T_m$ stands for the $m$-th column of $T$. Therefore,

$$\min_{1 \leq i \leq m-1} \delta_i = \min_{1 \leq j \leq m-1} \sum_{i=1}^{n} \mu_i^{-1} T_{ij} T_{im}.$$

□

## C.2 PROOF OF (6)

Let $T = e_\lambda[\alpha \oplus \beta - M]$, and then it is easy to find that $\nabla_\alpha \mathcal{L}(\alpha, \beta) = a - T\mathbf{1}_m$ and $\nabla_{\tilde{\beta}} \mathcal{L}(\alpha, \beta) = \tilde{b} - \tilde{T}^{\mathrm{T}}\mathbf{1}_n$. Since $\alpha^*(\beta) = \arg\max_\alpha \mathcal{L}(\alpha, \beta)$, we find that $\alpha_i \equiv \alpha^*(\beta)_i$ is the solution to the equation $a - T\mathbf{1}_m = 0$. By definition, we have

$$a_i = \sum_{j=1}^m e^{\lambda(\alpha_i + \beta_j - M_{ij})} = e^{\lambda\alpha_i} \sum_{j=1}^m e^{\lambda(\beta_j - M_{ij})}, \quad i = 1, \ldots, n,$$

so the solution is $\alpha_i = \eta \log a_i - \eta \log\left(\sum_{j=1}^m e^{\lambda(\beta_j - M_{ij})}\right)$. Since $T\mathbf{1}_m = a$, we immediately get $\mathbf{1}_n^{\mathrm{T}} T\mathbf{1}_m = 1$, so

$$\mathcal{L}(\alpha^*(\beta), \beta) = \alpha^*(\beta)^{\mathrm{T}} a + \beta^{\mathrm{T}} b - \eta \mathbf{1}_n^{\mathrm{T}} T\mathbf{1}_m = \alpha^*(\beta)^{\mathrm{T}} a + \beta^{\mathrm{T}} b - \eta,$$

and we get the expression for $f(\beta) = -\mathcal{L}(\alpha^*(\beta), \beta)$. Finally,

$$\nabla_{\tilde{\beta}} \mathcal{L}(\alpha^*(\beta), \beta) = \left[\frac{\partial \alpha^*(\beta)}{\partial \tilde{\beta}^{\mathrm{T}}}\right]^{\mathrm{T}} \nabla_\alpha \mathcal{L}(\alpha, \beta)\big|_{\alpha = \alpha^*(\beta)} + \nabla_{\tilde{\beta}} \mathcal{L}(\alpha, \beta)\big|_{\alpha = \alpha^*(\beta)}.$$

Since $\alpha^*(\beta) = \arg\max_\alpha \mathcal{L}(\alpha, \beta)$ implies that $\nabla_\alpha \mathcal{L}(\alpha, \beta)\big|_{\alpha = \alpha^*(\beta)} = 0$, we have $\nabla_{\tilde{\beta}} \mathcal{L}(\alpha^*(\beta), \beta) = \nabla_{\tilde{\beta}} \mathcal{L}(\alpha, \beta)\big|_{\alpha = \alpha^*(\beta)}$, and hence

$$\nabla_{\tilde{\beta}} f(\beta) = -\nabla_{\tilde{\beta}} \mathcal{L}(\alpha^*(\beta), \beta) = \tilde{T}(\beta)^{\mathrm{T}} \mathbf{1}_n - \tilde{b}.$$

## C.3 PROOF OF THEOREM 1

By definition we have

$$S_\lambda(M, a, b) = \langle T^*, M \rangle = \mathbf{vec}(T^*)^{\mathrm{T}} \mathbf{vec}(M),$$

so Lemma 2 gives

$$\frac{\partial S_\lambda(M, a, b)}{\partial \mathbf{vec}(M)^{\mathrm{T}}} = \mathbf{vec}(M)^{\mathrm{T}} \frac{\partial \mathbf{vec}(T^*)}{\partial \mathbf{vec}(M)^{\mathrm{T}}} + \mathbf{vec}(T^*)^{\mathrm{T}} \frac{\partial \mathbf{vec}(M)}{\partial \mathbf{vec}(M)^{\mathrm{T}}}. \tag{10}$$

Obviously, $\partial \mathbf{vec}(M)/\partial \mathbf{vec}(M)^{\mathrm{T}}$ is the $(nm) \times (nm)$ identity matrix $I_{(nm)}$, so the second term of (10) is essentially $\mathbf{vec}(T^*)^{\mathrm{T}}$, and the remaining task is to derive $\partial \mathbf{vec}(T^*)/\partial \mathbf{vec}(M)^{\mathrm{T}}$.

Let $R = \alpha^* \oplus \beta^* - M = \alpha^* \mathbf{1}_m^{\mathrm{T}} + \mathbf{1}_n \beta^{*\mathrm{T}} - M$, and then $T^* = e_\lambda[R]$. Using the chain rule of derivatives, we have

$$\frac{\partial \mathbf{vec}(T^*)}{\partial \mathbf{vec}(M)^{\mathrm{T}}} = \frac{\partial \mathbf{vec}(T^*)}{\partial \mathbf{vec}(R)^{\mathrm{T}}} \cdot \frac{\partial \mathbf{vec}(R)}{\partial \mathbf{vec}(M)^{\mathrm{T}}}. \tag{11}$$

It is easy to find that $\partial \mathbf{vec}(T^*)/\partial \mathbf{vec}(R)^{\mathrm{T}}$ is an $(nm) \times (nm)$ diagonal matrix with diagonal elements $\mathbf{vec}(\lambda T^*)$, so

$$\mathbf{vec}(M)^{\mathrm{T}} \frac{\partial \mathbf{vec}(T^*)}{\partial \mathbf{vec}(R)^{\mathrm{T}}} = \lambda \mathbf{vec}(M \odot T^*)^{\mathrm{T}}. \tag{12}$$

Furthermore,

$$\frac{\partial \mathbf{vec}(R)}{\partial \mathbf{vec}(M)^{\mathrm{T}}} = \frac{\partial \mathbf{vec}(\alpha^* \mathbf{1}_m^{\mathrm{T}} + \mathbf{1}_n \beta^{*\mathrm{T}} - M)}{\partial \mathbf{vec}(M)^{\mathrm{T}}}$$

$$= (\mathbf{1}_m \otimes I_n) \frac{\partial \alpha^*}{\partial \mathbf{vec}(M)^{\mathrm{T}}} + (I_m \otimes \mathbf{1}_n) \frac{\partial \beta^*}{\partial \mathbf{vec}(M)^{\mathrm{T}}} - I_{(nm)}, \tag{13}$$

where the second identity is an application of Lemma 3. Combine (11), (12) and (13), and then we get

$$\mathbf{vec}(M)^{\mathrm{T}} \frac{\partial \mathbf{vec}(T^*)}{\partial \mathbf{vec}(M)^{\mathrm{T}}} = \lambda \mathbf{vec}(M \odot T^*)^{\mathrm{T}} \frac{\partial \mathbf{vec}(R)}{\partial \mathbf{vec}(M)^{\mathrm{T}}}$$

$$= \lambda \mathbf{vec}(M \odot T^*)^{\mathrm{T}} (\mathbf{1}_m \otimes I_n) \frac{\partial \alpha^*}{\partial \mathbf{vec}(M)^{\mathrm{T}}}$$

$$+ \lambda \mathbf{vec}(M \odot T^*)^{\mathrm{T}} (I_m \otimes \mathbf{1}_n) \frac{\partial \beta^*}{\partial \mathbf{vec}(M)^{\mathrm{T}}}$$

$$- \lambda \mathbf{vec}(M \odot T^*)^{\mathrm{T}},$$

which is the first term of (10).

Using the identities in Lemma 1, we have

$$\mathbf{vec}(M \odot T^*)^{\mathrm{T}}(\mathbf{1}_m \otimes I_n) = \left[(\mathbf{1}_m \otimes I_n)^{\mathrm{T}}\mathbf{vec}(M \odot T^*)\right]^{\mathrm{T}} = \left[(\mathbf{1}_m^{\mathrm{T}} \otimes I_n)\mathbf{vec}(M \odot T^*)\right]^{\mathrm{T}}$$
$$= \left[\mathbf{vec}((M \odot T^*)\mathbf{1}_m)\right]^{\mathrm{T}} := \mu_r^{\mathrm{T}},$$

$$\mathbf{vec}(M \odot T^*)^{\mathrm{T}}(I_m \otimes \mathbf{1}_n) = \left[(I_m \otimes \mathbf{1}_n)^{\mathrm{T}}\mathbf{vec}(M \odot T^*)\right]^{\mathrm{T}} = \left[(I_m \otimes \mathbf{1}_n^{\mathrm{T}})\mathbf{vec}(M \odot T^*)\right]^{\mathrm{T}}$$
$$= \left[\mathbf{vec}(\mathbf{1}_n^{\mathrm{T}}(M \odot T^*))\right]^{\mathrm{T}} := \mu_c^{\mathrm{T}}.$$

Since we have set $\beta_m^* = 0$, (10) simplifies to

$$\frac{\partial S_\lambda(M, a, b)}{\partial \mathbf{vec}(M)^{\mathrm{T}}} = \lambda\left[\mu_r^{\mathrm{T}}\frac{\partial \alpha^*}{\partial \mathbf{vec}(M)^{\mathrm{T}}} + \tilde{\mu}_c^{\mathrm{T}}\frac{\partial \tilde{\beta}^*}{\partial \mathbf{vec}(M)^{\mathrm{T}}} - \mathbf{vec}(M \odot T^*)^{\mathrm{T}}\right] + \mathbf{vec}(T^*)^{\mathrm{T}}. \quad (14)$$

Let $w^* = (\alpha^{*\mathrm{T}}, \tilde{\beta}^{*\mathrm{T}})^{\mathrm{T}}$, and then the main challenge is to calculate $\partial w^*/\partial \mathbf{vec}(M)^{\mathrm{T}}$.

First, note that the optimality condition for $(\alpha^*, \beta^*) = \arg\max_{\alpha,\beta} \mathcal{L}(\alpha, \beta)$ is

$$\nabla_\alpha \mathcal{L}(\alpha, \beta)|_{(\alpha,\beta)=(\alpha^*,\beta^*)} = \mathbf{0}, \quad \nabla_\beta \mathcal{L}(\alpha, \beta)|_{(\alpha,\beta)=(\alpha^*,\beta^*)} = \mathbf{0}. \quad (15)$$

Section C.2 has shown that $\nabla_\alpha \mathcal{L}(\alpha, \beta) = a - T\mathbf{1}_m$ and $\nabla_{\tilde{\beta}}\mathcal{L}(\alpha, \beta) = \tilde{b} - \tilde{T}^{\mathrm{T}}\mathbf{1}_n$. Moreover,

$$\nabla_\alpha^2 \mathcal{L}(\alpha, \beta) = -\lambda\mathbf{diag}(T\mathbf{1}_m)$$
$$\nabla_{\tilde{\beta}}^2 \mathcal{L}(\alpha, \beta) = -\lambda\mathbf{diag}(\tilde{T}^{\mathrm{T}}\mathbf{1}_n)$$
$$\nabla_{\tilde{\beta}}\left(\nabla_\alpha \mathcal{L}(\alpha, \beta)\right) = -\lambda\tilde{T}.$$

Define the function

$$F(w, M) = \begin{pmatrix} \nabla_\alpha \mathcal{L}(\alpha, \beta) \\ \nabla_{\tilde{\beta}}\mathcal{L}(\alpha, \beta) \end{pmatrix} = \begin{pmatrix} a - T\mathbf{1}_m \\ \tilde{b} - \tilde{T}^{\mathrm{T}}\mathbf{1}_n \end{pmatrix}, \quad (16)$$

where $w = (\alpha^{\mathrm{T}}, \tilde{\beta}^{\mathrm{T}})^{\mathrm{T}}$, and then $\tilde{w}^*$ satisfies the equation $F(w^*, M) = \mathbf{0}$, indicating that $w^*$ is implicitly a function of $M$, written as $w^* = w(M)$. By the implicit function theorem,

$$\frac{\partial w(M)}{\partial \mathbf{vec}(M)^{\mathrm{T}}} = -\left[\frac{\partial F(w, M)}{\partial w^{\mathrm{T}}}\bigg|_{w=w^*}\right]^{-1}\frac{\partial F(w, M)}{\partial \mathbf{vec}(M)^{\mathrm{T}}}\bigg|_{w=w^*} := -F_w^{-1}F_M.$$

Note that

$$F_w = F_w^{\mathrm{T}} = \begin{pmatrix} \nabla_\alpha^2 \mathcal{L}(\alpha, \beta) & \nabla_{\tilde{\beta}}\left(\nabla_\alpha \mathcal{L}(\alpha, \beta)\right) \\ \left[\nabla_{\tilde{\beta}}\left(\nabla_\alpha \mathcal{L}(\alpha, \beta)\right)\right]^{\mathrm{T}} & \nabla_{\tilde{\beta}}^2 \mathcal{L}(\alpha, \beta) \end{pmatrix} := -\lambda\begin{pmatrix} A & \tilde{B} \\ \tilde{B}^{\mathrm{T}} & \tilde{D} \end{pmatrix}.$$

Then by the inversion formula for block matrices, we have

$$F_w^{-1} = -\lambda^{-1}\begin{pmatrix} A & \tilde{B} \\ \tilde{B}^{\mathrm{T}} & \tilde{D} \end{pmatrix}^{-1} = -\lambda^{-1}\begin{pmatrix} A^{-1} + A^{-1}\tilde{B}\tilde{\Delta}^{-1}\tilde{B}^{\mathrm{T}}A^{-1} & -A^{-1}\tilde{B}\tilde{\Delta}^{-1} \\ -\tilde{\Delta}^{-1}\tilde{B}^{\mathrm{T}}A^{-1} & \tilde{\Delta}^{-1} \end{pmatrix},$$

where $\tilde{\Delta} = \tilde{D} - \tilde{B}^{\mathrm{T}}A^{-1}\tilde{B}$. For $g = (\mu_r^{\mathrm{T}}, \tilde{\mu}_c^{\mathrm{T}})^{\mathrm{T}}$, the vector $\tilde{s} = (s_u^{\mathrm{T}}, \tilde{s}_v^{\mathrm{T}})^{\mathrm{T}} = -\lambda F_w^{-1}\tilde{g}$ has the following expression:

$$\tilde{s}_v = -\tilde{\Delta}^{-1}\tilde{B}^{\mathrm{T}}A^{-1}\mu_r + \tilde{\Delta}^{-1}\tilde{\mu}_c$$
$$s_u = A^{-1}\mu_r + A^{-1}\tilde{B}\tilde{\Delta}^{-1}\tilde{B}^{\mathrm{T}}A^{-1}\mu_r - A^{-1}\tilde{B}\tilde{\Delta}^{-1}\tilde{\mu}_c,$$
$$= A^{-1}\mu_r - A^{-1}\tilde{B}\tilde{s}_v.$$

After some simplification, we obtain

$$\tilde{\Delta} = \mathbf{diag}(\tilde{T}^{\mathrm{T}}\mathbf{1}_n) - \tilde{T}^{\mathrm{T}}\mathbf{diag}((T\mathbf{1}_m)^-)\tilde{T}$$
$$\tilde{s}_v = \tilde{\Delta}^{-1}\tilde{\mu}_c - \tilde{\Delta}^{-1}\tilde{T}^{\mathrm{T}}((T\mathbf{1}_m)^- \odot \mu_r)$$
$$s_u = (T\mathbf{1}_m)^- \odot \mu_r - (T\mathbf{1}_m)^- \odot (\tilde{T}\tilde{s}_v).$$

Next, partition $F_M$ as $F_M = \begin{pmatrix} G_M \\ \tilde{H}_M \end{pmatrix}$, where $G_M \in \mathbb{R}^{n \times (nm)}$ and $\tilde{H}_M \in \mathbb{R}^{(m-1) \times (nm)}$. By definition,

$$G_M = \begin{pmatrix} \frac{\partial G_1}{\partial \mathbf{vec}(M)^{\mathrm{T}}} \\ \vdots \\ \frac{\partial G_n}{\partial \mathbf{vec}(M)^{\mathrm{T}}} \end{pmatrix}, \quad G_i = -\sum_{j=1}^{m} T_{ij} = -\sum_{j=1}^{m} e^{\lambda(\alpha_i + \beta_j - M_{ij})},$$

so

$$\frac{\partial G_i}{\partial M_{kl}} = \begin{cases} 0, & i \neq k \\ \lambda T_{kl}, & i = k \end{cases}.$$

This indicates that $G_M = \lambda\left(\mathbf{diag}(T_1), \ldots, \mathbf{diag}(T_m)\right)$, where $T_1, \ldots, T_m$ are the column vectors of $T$. Similarly, for $H_j = -\sum_{i=1}^{n} T_{ij}$,

$$\tilde{H}_M = \begin{pmatrix} \frac{\partial H_1}{\partial \mathbf{vec}(M)^{\mathrm{T}}} \\ \vdots \\ \frac{\partial H_{m-1}}{\partial \mathbf{vec}(M)^{\mathrm{T}}} \end{pmatrix}, \quad \frac{\partial H_j}{\partial M_{kl}} = \begin{cases} 0, & j \neq l \\ \lambda T_{kl}, & j = l \end{cases},$$

implying that

$$\tilde{H}_M = \lambda \begin{pmatrix} T_1^{\mathrm{T}} & & \\ & \ddots & \\ & & T_{m-1}^{\mathrm{T}} \quad \mathbf{0}_n^{\mathrm{T}} \end{pmatrix}.$$

As a result,

$$\mu_r^{\mathrm{T}} \frac{\partial \alpha^*}{\partial \mathbf{vec}(M)^{\mathrm{T}}} + \tilde{\mu}_c^{\mathrm{T}} \frac{\partial \tilde{\beta}^*}{\partial \mathbf{vec}(M)^{\mathrm{T}}}$$

$$= (\mu_r^{\mathrm{T}}, \tilde{\mu}_c^{\mathrm{T}}) \frac{\partial w^*}{\partial \mathbf{vec}(M)^{\mathrm{T}}} = -(\mu_r^{\mathrm{T}}, \tilde{\mu}_c^{\mathrm{T}}) F_w^{-1} F_M$$

$$= \left[-\lambda F_w^{-1} \begin{pmatrix} \mu_r \\ \tilde{\mu}_c \end{pmatrix}\right]^{\mathrm{T}} \begin{pmatrix} \lambda^{-1} G_M \\ \lambda^{-1} \tilde{H}_M \end{pmatrix} = (s_u^{\mathrm{T}}, \tilde{s}_v^{\mathrm{T}}) \begin{pmatrix} \lambda^{-1} G_M \\ \lambda^{-1} \tilde{H}_M \end{pmatrix}$$

$$= \left((s_u \odot T_1)^{\mathrm{T}}, \ldots, (s_u \odot T_m)^{\mathrm{T}}\right) + \left(\tilde{s}_{v,1} T_1^{\mathrm{T}}, \ldots, \tilde{s}_{v,m-1} T_{m-1}^{\mathrm{T}}, \mathbf{0}_n^{\mathrm{T}}\right)$$

$$= \left[\mathbf{vec}\left(\mathbf{diag}(s_u) T + T \mathbf{diag}(s_v)\right)\right]^{\mathrm{T}}. \tag{17}$$

Finally, substitute (17) into (14), and we have

$$\frac{\partial S_\lambda(M, a, b)}{\partial \mathbf{vec}(M)^{\mathrm{T}}} = \lambda \left[\mathbf{vec}\left(\mathbf{diag}(s_u) T^* + T^* \mathbf{diag}(s_v)\right) - \mathbf{vec}(M \odot T)\right]^{\mathrm{T}} + \mathbf{vec}(T)^{\mathrm{T}}$$

Transforming back to the matrix form, and we obtain

$$\frac{\partial S_\lambda(M, a, b)}{\partial M} = \lambda(s_u \oplus s_v - M) \odot T + T.$$

Replacing $T$ with $T^*$ and noting that $a = T^* \mathbf{1}_m$, $\tilde{b} = \tilde{T}^{*\mathrm{T}} \mathbf{1}_n$, we get the stated result. The positive definiteness of the $\tilde{\Delta}$ matrix is a direct consequence of Lemma 8.

## C.4 PROOF OF THEOREM 2

In the proof of Theorem 1 we have already shown that

$$\nabla^2_{\alpha, \tilde{\beta}} \mathcal{L}(\alpha, \beta) = -\lambda H := -\lambda \begin{pmatrix} \mathbf{diag}(T \mathbf{1}_m) & \tilde{T} \\ \tilde{T}^{\mathrm{T}} & \mathbf{diag}(\tilde{T}^{\mathrm{T}} \mathbf{1}_n) \end{pmatrix},$$

where $T = \mathrm{e}_\lambda[\alpha \oplus \beta - M]$. Plugging $\alpha^*(\beta)$ into $\mathcal{L}(\alpha, \beta)$, and then $\nabla^2_{\tilde{\beta}} \mathcal{L}(\alpha^*(\beta), \beta)$ is the Schur complement of the top left block of $\nabla^2_{\alpha, \tilde{\beta}} \mathcal{L}(\alpha, \beta)$, given by

$$\nabla^2_{\tilde{\beta}} \mathcal{L}(\alpha^*(\beta), \beta) = -\lambda \left[\mathbf{diag}(\tilde{T}^{\mathrm{T}} \mathbf{1}_n) - \tilde{T}^{\mathrm{T}} \mathbf{diag}(T \mathbf{1}_m)^{-1} \tilde{T}\right].$$

Since $f(\beta) = -\mathcal{L}(\alpha^*(\beta), \beta)$, by Lemma 8 we find that $\nabla_{\tilde{\beta}}^2 f(\beta)$ is positive definite, so $f(\beta)$ is strictly convex on $\tilde{\beta}$, and hence $\beta^*$ is unique.

The optimality conditions for $(\alpha^*, \beta^*)$ are $T^* \mathbf{1}_m = a$ and $T^{*\mathrm{T}} \mathbf{1}_n = b$, where $T^* = \mathrm{e}_\lambda[\alpha^* \oplus \beta^* - M]$. Since $T_{ij}^* = \exp\{\lambda(\alpha_i^* + \beta_j^* - M_{ij})\} \geq 0$ and $a_i = \sum_{j=1}^m T_{ij}^*$, $b_j = \sum_{i=1}^n T_{ij}^*$, we have $T_{ij}^* \leq \min\{a_i, b_j\}$ for all $i$ and $j$, implying that

$$\alpha_i^* + \beta_j^* \leq U_{ij} := M_{ij} + \lambda^{-1} \min\{\log(a_i), \log(b_j)\}.$$

Since $\beta_m^* = 0$ by design, we have $\alpha_i^* \leq U_{\alpha_i} < +\infty, i = 1, \ldots, n$, where $U_{\alpha_i} = U_{im} = M_{im} + \lambda^{-1} \min\{\log(a_i), \log(b_m)\}$. This indicates that $\alpha_i^*$ is upper bounded.

Next, let $I = \arg\max_i T_{im}^*$. Since $T^{*\mathrm{T}} \mathbf{1}_n = b$ implies that $b_m = \sum_{i=1}^n T_{im}^* \leq n T_{Im}^*$, we have
$$T_{Im}^* = \exp\{\lambda(\alpha_I^* - M_{Im})\} \geq b_m/n,$$
and hence $\alpha_I^* \geq M_{Im} + \lambda^{-1} \log(b_m/n)$. Again, since $\alpha_i^* + \beta_j^* \leq U_{ij}$ for all $i$ and $j$, it holds that

$$\begin{aligned}
\beta_j^* &\leq U_{Ij} - \alpha_I^* \leq U_{Ij} - M_{Im} - \lambda^{-1} \log(b_m/n) \\
&= M_{Ij} + \lambda^{-1} \min\{\log(a_I), \log(b_j)\} - M_{Im} - \lambda^{-1} \log(b_m/n) \\
&= M_{Ij} - M_{Im} + \lambda^{-1} \min\{\log(na_I/b_m), \log(nb_j/b_m)\} \\
&:= U_{\beta_j} < +\infty, \quad j = 1, \ldots, m.
\end{aligned} \tag{18}$$

On the other hand, $T^{*\mathrm{T}} \mathbf{1}_n = b$ implies that $b_j = \sum_{i=1}^n T_{ij}^* = e^{\lambda \beta_j^*} \cdot \sum_{i=1}^n e^{\lambda(\alpha_i^* - M_{ij})}$ for any $j$, so

$$\log b_j = \lambda \beta_j^* + \log\left[\sum_{i=1}^n e^{\lambda(\alpha_i^* - M_{ij})}\right] \leq \lambda \beta_j^* + \log\left[\sum_{i=1}^n e^{\lambda(U_{\alpha_i} - M_{ij})}\right].$$

It is well-known that

$$\max\{x_1, \ldots, x_n\} \leq \log\left(\sum_{i=1}^n e^{x_i}\right) \leq \max\{x_1, \ldots, x_n\} + \log n$$

for any $x_1, \ldots, x_n \in \mathbb{R}$, so

$$\begin{aligned}
\beta_j^* &\geq \lambda^{-1} \log b_j - \lambda^{-1} \log\left[\sum_{i=1}^n e^{\lambda(U_{\alpha_i} - M_{ij})}\right] \\
&\geq \lambda^{-1} \log b_j - \max_i (U_{\alpha_i} - M_{ij}) - \lambda^{-1} \log n \\
&\geq \lambda^{-1} \log(b_j/n) - \max_i (U_{\alpha_i} - M_{ij}) := L_{\beta_j} > -\infty, \quad j = 1, \ldots, m.
\end{aligned} \tag{19}$$

Then (18) and (19) together show that $|\beta_j^*| < \infty$.

Similarly, $T^* \mathbf{1}_m = a$, so

$$\log a_i \leq \lambda \alpha_i^* + \log\left[\sum_{j=1}^m e^{\lambda(U_{\beta_j} - M_{ij})}\right],$$

$$\alpha_i^* \geq \lambda^{-1} \log(a_i/m) - \max_j (U_{\beta_j} - M_{ij}) := L_{\alpha_i} > -\infty, \quad i = 1, \ldots, n.$$

The trivial bounds $\underline{L}_{\alpha_i}$ and $\overline{U}_{\beta_j}$ are obtained by removing the unknown index $I$.

The results above verify that $|\alpha_i^*| < \infty$ and $|\beta_j^*| < \infty$, and hence $\|\alpha^*\| < \infty$ and $\|\beta^*\| < \infty$. Finally, plugging in $\beta^*$ to the objective function, and we immediately get $f^* > -\infty$.

## C.5  PROOF OF THEOREM 3

Claims (a) and (b) are direct consequences of the convergence property of the L-BFGS algorithm (Theorem 7.1, Liu & Nocedal, 1989), and we only need to verify its three assumptions. The new results here are explicit expressions for the constants $C_1$, $C_2$, and $r$.

First, $f$ is twice continuously differentiable, so Assumption 7.1(1) of Liu & Nocedal (1989) is verified. Second, $f$ is a closed convex function, and we define the level set of $f$ as $L_c = \{\tilde{\beta} \in \mathbb{R}^{m-1} : f(\beta) \leq c\}$. Theorem 2 has shown that $f^* > -\infty$, and when $c = f^*$, obviously $L_c = \{\tilde{\beta}^*\}$ is non-empty and bounded. Then Corollary 8.7.1 of Rockafellar (1970) shows that $L_c$ is bounded for every $c$. In particular, for a fixed initial value $\tilde{\beta}^{(0)}$, define $L = \{\tilde{\beta} : f(\beta) \leq f(\beta^{(0)})\}$, and then $L$ is a bounded, closed, and convex set, which verifies Assumption 7.1(2) of Liu & Nocedal (1989). Third, let $H(\beta) := \nabla^2_{\tilde{\beta}} f(\beta)$, and then in the proof of Theorem 2 we have already shown that

$$H(\beta) = \lambda \left[ \mathbf{diag}(\tilde{T}^{\mathrm{T}} \mathbf{1}_n) - \tilde{T}^{\mathrm{T}} \mathbf{diag}(T\mathbf{1}_m)^{-1} \tilde{T} \right],$$

where $T = \mathrm{e}_\lambda[\alpha^*(\beta) \oplus \beta - M]$. Lemma 8 verifies that

$$\sigma_{\min}(H(\beta)) \geq \lambda \cdot \frac{n - m + 2}{2(n - m + 1)} \cdot \min_{1 \leq i \leq n} T_{im}, \qquad \sigma_{\max}(H(\beta)) \leq \lambda \cdot \max_{1 \leq j \leq m-1} \nu_j,$$

where $\nu = T^{\mathrm{T}} \mathbf{1}_n$. On the $L$ set, Lemma 7 shows that $\max_j \beta_j \leq U_c$ and $\min_j \beta_j \geq L_c$, with $c = f(\beta^{(0)})$. Therefore,

$$\alpha_i := \eta \log a_i - \eta \log \left[ \sum_{j=1}^m e^{\lambda(\beta_j - M_{ij})} \right] \geq \eta \log a_i - \eta \log \left[ e^{-\lambda M_{im}} + \sum_{j=1}^{m-1} e^{\lambda(U_c - M_{ij})} \right]$$

$$= \eta \log a_i - U_c - \eta \log \left( e^{-\lambda(M_{im} + U_c)} + \sum_{j=1}^{m-1} e^{-\lambda M_{ij}} \right) := A_i,$$

$$T_{im} = e^{\lambda(\alpha_i - M_{im})} \geq e^{\lambda(A_i - M_{im})}.$$

On the other hand, $T_{ij}$ must satisfy $T_{ij} > 0$ and $\sum_{j=1}^m T_{ij} = a_i$ for any $i$ and $j$, so for $j = 1, \ldots, m-1$, we have $T_{ij} \leq a_i - T_{im}$. Therefore,

$$v_j = \sum_{i=1}^n T_{ij} \leq \sum_{i=1}^n (a_i - T_{im}) = 1 - \sum_{i=1}^n T_{im} \leq 1 - \sum_{i=1}^n e^{\lambda(A_i - M_{im})}, \quad j = 1, \ldots, m-1.$$

This implies that there exist constants $M_1, M_2 > 0$ such that

$$M_1 \|x\|^2 \leq x^{\mathrm{T}} H(\beta) x \leq M_2 \|x\|^2$$

for all $x \in \mathbb{R}^{m-1}$ and $\tilde{\beta} \in L$, with

$$M_1 = \lambda \cdot \frac{n - m + 2}{2(n - m + 1)} \cdot \min_{1 \leq i \leq n} e^{\lambda(A_i - M_{im})},$$

$$M_2 = \lambda \left[ 1 - \sum_{i=1}^n e^{\lambda(A_i - M_{im})} \right].$$

This verifies Assumption 7.1(3) of Liu & Nocedal (1989).

Next, we derive the explicit constants in the theorem. Following the notations in equation (7.3) of Liu & Nocedal (1989), the BFGS matrix $B_k$ for the L-BFGS algorithm has the following expression

$$B_k = B^{(\tilde{m})}, \quad B^{(l+1)} = B^{(l)} - \frac{B^{(l)} s_l s_l^{\mathrm{T}} B^{(l)}}{s_l^{\mathrm{T}} B^{(l)} s_l} + \frac{y_l y_l^{\mathrm{T}}}{y_l^{\mathrm{T}} s_l},$$

where $\tilde{m} = \min\{k+1, m_0\}$, $m_0$ is a user-defined constant explained in Section A, and $\{y_l\}$ and $\{s_l\}$ are two sequences of vectors. We also choose $B^{(0)} = I$ to be the identity matrix.

Fix $l = \tilde{m}$, and let $\cos \theta_k = s_l^{\mathrm{T}} B^{(l)} s_l / (\|s_l\| \cdot \|B^{(l)} s_l\|)$, $\rho_k = y_l^{\mathrm{T}} s_l / \|s_l\|^2$, $\tau_k = \|y_l\|^2 / y_l^{\mathrm{T}} s_l$, and $q_k = s_l^{\mathrm{T}} B^{(l)} s_l / \|s_l\|^2$. Then it can be verified that

$$\mathrm{tr}(B^{(l+1)}) = \mathrm{tr}(B^{(l)}) - \frac{\|B^{(l)} s_l\|^2}{s_l^{\mathrm{T}} B^{(l)} s_l} + \frac{\|y_l\|^2}{y_l^{\mathrm{T}} s_l} = \mathrm{tr}(B^{(l)}) - \frac{q_k}{\cos^2 \theta_k} + \tau_k, \qquad (20)$$

$$\det(B^{(l+1)}) = \det(B^{(l)}) \rho_k / q_k. \qquad (21)$$

Define $\psi(B) = \mathrm{tr}(B) - \log\det(B)$, and it is known that $\psi(B) > 0$ for any positive definite matrix $B$. Equation (6.50) of Nocedal & Wright (2006) shows that

$$0 < \psi(B^{(l+1)}) = \mathrm{tr}(B^{(l)}) - \frac{q_k}{\cos^2\theta_k} + \tau_k - \log\det(B^{(l)}) - \log\rho_k + \log q_k$$

$$= \psi(B^{(l)}) + (\tau_k - \log\rho_k - 1) + \left(1 - \frac{q_k}{\cos^2\theta_k} + \log\frac{q_k}{\cos^2\theta_k}\right) + \log\cos^2\theta_k.$$

Under the assumptions verified above, equations (7.8) and (7.9) of Liu & Nocedal (1989) show that $M_1 \leq y_l^{\mathrm{T}} s_l / \|s_l\|^2 \leq M_2$ and $\|y_l\|^2 / y_l^{\mathrm{T}} s_l \leq M_2$ for every $l$, so $M_1 \leq \rho_k \leq M_2$ and $\tau_k \leq M_2$. Since $h(x) = 1 - x + \log(x) \leq 0$ for all $x > 0$, we have

$$0 < \psi(B_k) + (M_2 - \log M_1 - 1) + \log\cos^2\theta_k := \psi(B_k) + M_3 + \log\cos^2\theta_k. \tag{22}$$

Now we show that $\psi(B_k)$ can be upper bounded. First, Lemma 5 implies that for $l = 0, \ldots, \tilde{m} - 1$,

$$\sigma_{\max}(B^{(l+1)}) \leq \sigma_{\max}(B^{(l)}) + 0 + \|y_l\|^2 / y_l^{\mathrm{T}} s_l \leq \sigma_{\max}(B^{(l)}) + M_2,$$

so $\sigma_{\max}(B_k) \leq 1 + m_0 M_2$. This also implies that $q_k \leq 1 + m_0 M_2$. Next, (20) shows that $\mathrm{tr}(B_k) \leq m - 1 + m_0 M_2$, and (21) gives

$$\log\det(B^{(l+1)}) = \log\det(B^{(l)}) + \log\rho_k - \log q_k \geq \log\det(B^{(l)}) + \log M_1 - \log(1 + m_0 M_2),$$

implying that $\log\det(B_k) \geq m_0 [\log M_1 - \log(1 + m_0 M_2)]$. As a result, we get

$$\psi(B_k) \leq M_4 := m - 1 + m_0 M_2 - m_0 [\log M_1 - \log(1 + m_0 M_2)]. \tag{23}$$

Combining (22) and (23), we have $\cos^2\theta_k > e^{-(M_3 + M_4)}$.

Finally, using the argument in Byrd et al. (1987), we have $f^{(k+1)} - f^* \leq r(f^{(k)} - f^*)$, where

$$r = 1 - c_1(1 - c_2) M_1 / M_2 \cos^2\theta_k < 1 - c_1(1 - c_2) M_1 / M_2 e^{-(M_3 + M_4)},$$

and $c_1, c_2$ are two constants for the Wolfe condition as explained in Section A. The constant $C_1$ is simply $2/M_1$.

For (c), we follow the analysis in Nocedal et al. (2002). Let $g(\beta) = \nabla_{\tilde{\beta}} f(\beta)$, and then $g(\beta^*) = \mathbf{0}$. By Taylor's theorem we have

$$f(\beta) - f^* = \frac{1}{2}(\tilde{\beta} - \tilde{\beta}^*)^{\mathrm{T}} H_1 (\tilde{\beta} - \tilde{\beta}^*), \tag{24}$$

where $H_1 = H(\xi)$ for some $\xi$ in the line segment connecting $\tilde{\beta}$ and $\tilde{\beta}^*$. Also,

$$g(\beta) - g(\beta^*) = g(\beta) = H_2(\tilde{\beta} - \tilde{\beta}^*), \quad H_2 = \int_0^1 H(\tilde{\beta} + t(\tilde{\beta}^* - \tilde{\beta}))\mathrm{d}t. \tag{25}$$

Combining (24) and (25), we get

$$\|g(\beta)\|^2 = (\tilde{\beta} - \tilde{\beta}^*)^{\mathrm{T}} H_2^2 (\tilde{\beta} - \tilde{\beta}^*) = \frac{2(\tilde{\beta} - \tilde{\beta}^*)^{\mathrm{T}} H_2^2 (\tilde{\beta} - \tilde{\beta}^*)}{(\tilde{\beta} - \tilde{\beta}^*)^{\mathrm{T}} H_1 (\tilde{\beta} - \tilde{\beta}^*)} \cdot (f(\beta) - f^*)$$

for any $\tilde{\beta} \in L$. Let $x = H_1^{1/2}(\tilde{\beta} - \tilde{\beta}^*)$, and then

$$\|g(\beta)\|^2 = \frac{2x^{\mathrm{T}} H_1^{-1/2} H_2^2 H_1^{-1/2} x}{\|x\|^2} \cdot (f(\beta) - f^*).$$

It is easy to find that

$$\sigma_{\max}(H_1^{-1}) = [\sigma_{\min}(H_1)]^{-1} \leq M_1^{-1}, \quad \sigma_{\max}(H_2^2) = [\sigma_{\max}(H_2)]^2 \leq M_2^2.$$

By Lemma 4, we have $x^{\mathrm{T}} H_1^{-1/2} H_2^2 H_1^{-1/2} x \leq M_1^{-1} M_2^2 \|x\|^2$ for any $x \in \mathbb{R}^{m-1}$, and hence $\|g(\beta)\|^2 \leq C_2(f(\beta) - f^*)$ for all $\tilde{\beta} \in L$, where $C_2 = 2M_1^{-1} M_2^2$. Since $f^{(k)} \leq f^{(0)}$ due to claim (a), we find that $\tilde{\beta}^{(k)} \in L$ for all $k > 0$. Therefore,

$$\|g(\gamma^{(k)})\|^2 \leq C_2(f^{(k)} - f^*) \leq C_2 r^k (f^{(0)} - f^*),$$

and claim (c) is proved.

Claims (d) and (e) can be verified as follows. For any $\tilde{\beta} \in L$, recall that $T = e_\lambda[\alpha^*(\beta) \oplus \beta - M]$, and then $T\mathbf{1}_m - a = 0$ and

$$\|\tilde{T}^{\mathrm{T}}\mathbf{1}_n - \tilde{b}\|_\infty \leq \|\tilde{T}^{\mathrm{T}}\mathbf{1}_n - \tilde{b}\| = \|\nabla_{\tilde{\beta}} f(\beta)\|,$$

indicating that

$$\left|\sum_{i=1}^{n} T_{ij} - b_j\right| \leq \|\nabla_{\tilde{\beta}} f(\beta)\|, \quad j = 1, \ldots, m-1. \tag{26}$$

Then $0 < T_{ij} \leq \sum_{j=1}^{m} T_{ij} = a_i$ and $0 < T_{ij} \leq \sum_{i=1}^{n} T_{ij} \leq b_j + \|\nabla_{\tilde{\beta}} f(\beta)\|$. The gradient $\|\nabla_{\tilde{\beta}} f(\beta)\|$ can be bounded using claim (c), so (d) is also proved. On the other hand, (26) shows that

$$a_i = \sum_{j=1}^{m} T_{ij} \leq m \cdot \max_j T_{ij},$$

and similarly we have

$$b_j - \|\nabla_{\tilde{\beta}} f(\beta)\| \leq \sum_{i=1}^{n} T_{ij} \leq n \cdot \max_i T_{ij}.$$

Replacing $\|\nabla_{\tilde{\beta}} f(\beta)\|$ by its upper bound, and claim (e) is verified.

### C.6  PROOF OF THEOREM 4

For matrix $A_{n \times m} = (a_{ij})$, define $\|A\|_\infty = \max_{i,j} |a_{ij}|$, and the notation $A \geq 0$ means $a_{ij} \geq 0$ for all $i$ and $j$. First, it is easy to show that $\|A \odot B\|_F \leq \|A\|_\infty \|B\|_F$, since

$$\|A \odot B\|_F = \sqrt{\sum_{i,j} (a_{ij} b_{ij})^2} \leq \sqrt{\sum_{i,j} \|A\|_\infty^2 b_{ij}^2} = \|A\|_\infty \|B\|_F.$$

Next, we show that if $B_{n \times m} \geq 0$, $C_{p \times n} \geq 0$, and $v \geq 0$, where $v$ is a vector, then $\|C(A \odot B)v\| \leq \|A\|_\infty \|CBv\|$.

*Proof.* Let $u_i$ be the $i$-th element of $(A \odot B)v$, and then

$$u_i = \sum_{j=1}^{m} a_{ij} b_{ij} v_j, \quad |u_i| \leq \|A\|_\infty \sum_{j=1}^{m} b_{ij} v_j.$$

Consequently,

$$\|C(A \odot B)v\| = \sqrt{\sum_{k=1}^{p} \left|\sum_{i=1}^{n} c_{ki} u_i\right|^2} \leq \sqrt{\sum_{k=1}^{p} \left(\sum_{i=1}^{n} c_{ki} |u_i|\right)^2} \leq \|A\|_\infty \sqrt{\sum_{k=1}^{p} \left[\sum_{i=1}^{n} c_{ki} \left(\sum_{j=1}^{m} b_{ij} v_j\right)\right]^2}$$

$$= \|A\|_\infty \|CBv\|.$$

$\square$

Similarly, if $A \geq 0$ and $v \geq 0$, then

$$\|A(u \odot v)\| = \sqrt{\sum_{i=1}^{n} \left|\sum_{j=1}^{m} a_{ij} u_j v_j\right|^2} \leq \|u\|_\infty \sqrt{\sum_{i=1}^{n} \left(\sum_{j=1}^{m} a_{ij} v_j\right)^2} = \|u\|_\infty \|Av\|.$$

Moreover, for matrices $B_{n \times m} \geq 0$, and $C_{m \times p} \geq 0$, let $C_j$ be the $j$-th column of $C$, and then

$$\|(A \odot B)C\|_F = \sqrt{\sum_{j=1}^{p} \|(A \odot B)C_i\|^2} \leq \|A\|_\infty \sqrt{\sum_{j=1}^{p} \|BC_i\|^2} = \|A\|_\infty \|BC\|_F.$$

Let $\alpha = \alpha^* + f$ and $\beta = \beta^* + g$ for some perturbation vectors $f$ and $g$, and define $T = \mathrm{e}_\lambda[\alpha \oplus \beta - M]$. Then it is easy to find that

$$T - T^* = \mathrm{e}_\lambda[\alpha^* \oplus \beta^* - M + (f \oplus g)] - T^* = \mathrm{e}_\lambda[f \oplus g] \odot T^* - T^*.$$

Let $E_T = \mathrm{e}_\lambda[f \oplus g] - \mathbf{1}_n \mathbf{1}_m'$, so $T - T^* = E_T \odot T^*$. Since $|e^x - 1| < 2|x|$ for $|x| < 1$, we have

$$|(E_T)_{ij}| = |e^{\lambda(f_i + g_j)} - 1| < 2\lambda|f_i + g_j|$$

as long as $\lambda|f_i + g_j| < 1$. This can be achieved by assuming $\varepsilon := 2\lambda(\|f\|_\infty + \|g\|_\infty) < 1$, since in this case $\lambda|f_i + g_j| \le \lambda(\|f\|_\infty + \|g\|_\infty) < 1/2$. Then clearly $\|E_T\|_\infty < \varepsilon$.

Consider $\hat{s}_u = s_u + \delta_u$ and $\hat{s}_v = s_v + \delta_v$ for some perturbation vectors $\delta_u$ and $\delta_v$, and let

$$\begin{aligned}
\widehat{\nabla_M S} &= T + \lambda(\hat{s}_u \oplus \hat{s}_v - M) \odot T \\
&= T + \lambda(s_u \oplus s_v - M) \odot T + \lambda(\delta_u \oplus \delta_v) \odot T \\
&= \nabla_M S + E_T \odot \nabla_M S + \lambda(\delta_u \oplus \delta_v) \odot (T^* + E_T \odot T^*).
\end{aligned}$$

Then we have

$$\begin{aligned}
\|\widehat{\nabla_M S} - \nabla_M S\|_F &\le \|E_T \odot \nabla_M S\|_F + \|\lambda(\delta_u \oplus \delta_v) \odot (T^* + E_T \odot T^*)\|_F \\
&\le \|E_T\|_\infty \|\nabla_M S\|_F + \lambda\|\delta_u \oplus \delta_v\|_\infty \|T^* + E_T \odot T^*\|_F \\
&< \varepsilon\|\nabla_M S\|_F + \lambda\|\delta_u \oplus \delta_v\|_\infty \|\mathbf{1}_n \mathbf{1}_m' + E_T\|_\infty \|T^*\|_F \\
&\le \varepsilon\|\nabla_M S\|_F + \lambda\|\delta_u \oplus \delta_v\|_\infty (1 + \|E_T\|_\infty)\|T^*\|_F \\
&< \varepsilon\|\nabla_M S\|_F + \lambda(1 + \varepsilon)\|\delta_u \oplus \delta_v\|_\infty \|T^*\|_F \\
&< \varepsilon\|\nabla_M S\|_F + 2\lambda\|T^*\|_F (\|\delta_u\|_\infty + \|\delta_v\|_\infty).
\end{aligned}$$

Therefore, we just need to show proper bounds for $\|\delta_u\|_\infty$ and $\|\delta_v\|_\infty$.

Consider $\hat{\mu}_r = (M \odot T)\mathbf{1}_m$ and $\hat{\mu}_c = (M \odot T)^{\mathrm{T}}\mathbf{1}_n$, and then

$$\begin{aligned}
\hat{\mu}_r &= (M \odot (T^* + E_T \odot T^*))\mathbf{1}_m = \mu_r + (E_T \odot M \odot T^*)\mathbf{1}_m := \mu_r + \delta_r, \\
\hat{\mu}_c &= (M \odot (T^* + E_T \odot T^*))^{\mathrm{T}}\mathbf{1}_n = \mu_c + (E_T \odot M \odot T^*)^{\mathrm{T}}\mathbf{1}_m := \mu_c + \delta_c.
\end{aligned}$$

It can be easily verified that

$$\begin{aligned}
\|a^- \odot \delta_r\| &= \|\mathbf{diag}(a^-)(E_T \odot M \odot T^*)\mathbf{1}_m\| \le \|E_T\|_\infty \|\mathbf{diag}(a^-)(M \odot T^*)\mathbf{1}_m\| < \varepsilon\|\mu_r\|, \\
\|T^{*\mathrm{T}}(a^- \odot \delta_r)\| &= \|T^{*\mathrm{T}}\mathbf{diag}(a^-)(E_T \odot M \odot T^*)\mathbf{1}_m\| \le \|E_T\|_\infty \|T^{*\mathrm{T}}\mathbf{diag}(a^-)(M \odot T^*)\mathbf{1}_m\| \\
&< \varepsilon\|T^{*\mathrm{T}}(a^- \odot \mu_r)\|, \\
\|\delta_c\| &= \|(E_T \odot M \odot T^*)^{\mathrm{T}}\mathbf{1}_m\| \le \|E_T\|_\infty \|(M \odot T^*)^{\mathrm{T}}\mathbf{1}_m\| < \varepsilon\|\mu_c\|.
\end{aligned}$$

Define $b_v = \mu_c - T^{*\mathrm{T}}(a^- \odot \mu_r)$ and $\hat{b}_v = \hat{\mu}_c - T^{\mathrm{T}}(a^- \odot \hat{\mu}_r)$, and we have

$$\begin{aligned}
\hat{b}_v &= \mu_c + \delta_c - (T^* + E_T \odot T^*)^{\mathrm{T}}(a^- \odot (\mu_r + \delta_r)) \\
&= \mu_c + \delta_c - T^{*\mathrm{T}}(a^- \odot (\mu_r + \delta_r)) - (E_T \odot T^*)^{\mathrm{T}}(a^- \odot (\mu_r + \delta_r)) \\
&= \mu_c - T^{*\mathrm{T}}(a^- \odot \mu_r) + \delta_c - T^{*\mathrm{T}}(a^- \odot \delta_r) - (E_T \odot T^*)^{\mathrm{T}}(a^- \odot (\mu_r + \delta_r))
\end{aligned}$$

As a result,

$$\begin{aligned}
\|\hat{b}_v - b_v\| &= \|\delta_c - T^{*\mathrm{T}}(a^- \odot \delta_r) - (E_T \odot T^*)^{\mathrm{T}}(a^- \odot (\mu_r + \delta_r))\| \\
&\le \|\delta_c\| + \|T^{*\mathrm{T}}(a^- \odot \delta_r)\| + \|E_T\|_\infty \|T^{*\mathrm{T}}(a^- \odot (\mu_r + \delta_r))\| \\
&< \varepsilon\|\mu_c\| + \varepsilon\|T^{*\mathrm{T}}(a^- \odot \mu_r)\| + \varepsilon\|T^{*\mathrm{T}}(a^- \odot \mu_r)\| + \varepsilon\|T^{*\mathrm{T}}(a^- \odot \delta_r)\| \\
&< \varepsilon\|\mu_c\| + \varepsilon\|T^{*\mathrm{T}}(a^- \odot \mu_r)\| + \varepsilon\|T^{*\mathrm{T}}(a^- \odot \mu_r)\| + \varepsilon^2\|T^{*\mathrm{T}}(a^- \odot \mu_r)\| \\
&< \varepsilon\|\mu_c\| + 3\varepsilon\|T^{*\mathrm{T}}(a^- \odot \mu_r)\|.
\end{aligned}$$

On the other hand, let $\hat{t}_{ij}$ be the $(i,j)$ element of the matrix $\tilde{T}^{\mathrm{T}}\mathbf{diag}(a^-)\tilde{T}$, and $t_{ij}$ be the $(i,j)$ element of $\tilde{T}^{*\mathrm{T}}\mathbf{diag}(a^-)\tilde{T}^*$. Then

$$\hat{t}_{ij} = \sum_{k=1}^{n} T_{ki}T_{kj}/a_k = \sum_{k=1}^{n} [1 + (E_T)_{ki}][1 + (E_T)_{kj}]T_{ki}^* T_{kj}^*/a_k,$$

$$|\hat{t}_{ij} - t_{ij}| = \left| \sum_{k=1}^{n} [(E_T)_{ki} + (E_T)_{kj} + (E_T)_{ki}(E_T)_{kj}]T_{ki}^* T_{kj}^*/a_k \right|$$

$$\leq (2\varepsilon + \varepsilon^2) \sum_{k=1}^{n} \left| T_{ki}^* T_{kj}^*/a_k \right| < 3\varepsilon t_{ij}.$$

This implies that

$$\|\hat{D} - D\|_F = \|\tilde{T}^{*\mathrm{T}}\mathbf{diag}(a^-)\tilde{T}^* - \tilde{T}^{\mathrm{T}}\mathbf{diag}(a^-)\tilde{T}\|_F = \sqrt{\sum_{i,j} |\hat{t}_{ij} - t_{ij}|^2} \leq 3\varepsilon\|D\|_F.$$

Then by Theorem 7.2 of Higham (2002), if $3\varepsilon\sigma\|D\|_F < 1$, we have

$$\frac{\|\delta_v\|}{\|s_v\|} \leq \frac{\varepsilon\sigma}{1 - 3\varepsilon\sigma\|D\|_F} \left( \frac{\|\mu_c\| + 3\|T^{*\mathrm{T}}(a^- \odot \mu_r)\|}{\|s_v\|} + 3\|D\|_F \right).$$

where $\sigma = \|D^{-1}\|_{\mathrm{op}} = 1/\sigma_{\min}(D)$. Assume that $\varepsilon < \min\{1, 1/(6\sigma\|D\|_F)\}$, then with slight simplification, we have $\|\delta_v\|_\infty \leq \|\delta_v\| \leq C_v\varepsilon$, where

$$C_v = 2\sigma(\|\mu_c\| + 3\|T^{*\mathrm{T}}(a^- \odot \mu_r)\| + 3\|D\|_F\|s_v\|).$$

On the other hand,

$$\begin{aligned}
\|\delta_u\|_\infty \leq \|\delta_u\| &\leq \|a^- \odot \delta_r\| + \|a^- \odot (T\hat{s}_v - T^* s_v)\| \\
&\leq \varepsilon\|\mu_r\| + \|a^- \odot (T^*\hat{s}_v + (E_T \odot T^*)\hat{s}_v - T^* s_v)\| \\
&\leq \varepsilon\|\mu_r\| + \|a^- \odot (T^*\delta_v)\| + \|a^- \odot ((E_T \odot T^*)\hat{s}_v)\| \\
&\leq \varepsilon\|\mu_r\| + \|a^- \odot (T^*\delta_v)\| + \|\mathbf{diag}(a^-)(E_T \odot T^*)\hat{s}_v\| \\
&\leq \varepsilon\|\mu_r\| + \|a^- \odot (T^*\delta_v)\| + \varepsilon\|\mathbf{diag}(a^-)T^*\hat{s}_v\| \\
&= \varepsilon\|\mu_r\| + \|a^- \odot (T^*\delta_v)\| + \varepsilon\|\mathbf{diag}(a^-)T^*(s_v + \delta_v)\| \\
&\leq \varepsilon\|\mu_r\| + (1+\varepsilon)\|a^- \odot (T^*\delta_v)\| + \varepsilon\|\mathbf{diag}(a^-)T^* s_v\| \\
&\leq \varepsilon\|\mu_r\| + (1+\varepsilon)\|\mathbf{diag}(a^-)T^*\|_F\|\delta_v\| + \varepsilon\|a^- \odot (T^* s_v)\| \\
&\leq \varepsilon(\|\mu_r\| + 2C_v\|\mathbf{diag}(a^-)T^*\|_F + \|a^- \odot (T^* s_v)\|).
\end{aligned}$$

Combining the results together, we get

$$\|\widehat{\nabla_M S} - \nabla_M S\|_F \leq \varepsilon \left[ \|\nabla_M S\|_F + 2\lambda\|T^*\|_F(C_v + C_u) \right],$$

where $C_u = \|\mu_r\| + 2C_v\|\mathbf{diag}(a^-)T^*\|_F + \|a^- \odot (T^* s_v)\|$.

Finally, Theorem 3(b) shows that $\|g\|_\infty \leq \|g\| = \|\beta^{(k)} - \beta^*\| \leq \sqrt{C_1\varepsilon^{(k)}}$. In addition, by (6) we have

$$\alpha_i^{(k)} := \alpha^*(\beta^{(k)})_i = \eta \log a_i - \eta \log \left[ \sum_{j=1}^{m} e^{\lambda(\beta_j^{(k)} - M_{ij})} \right].$$

Then using Lemma (6), we obtain

$$|\alpha_i^{(k)} - \alpha^*| = \eta \left| \log \left[ \sum_{j=1}^{m} e^{\lambda(\beta_j^{(k)} - M_{ij})} \right] - \log \left[ \sum_{j=1}^{m} e^{\lambda(\beta_j^* - M_{ij})} \right] \right| \leq \eta\|\lambda(\beta_j^{(k)} - \beta_j^*)\|_\infty = \|g\|_\infty,$$

implying that $\|f\|_\infty = \|\alpha^*(\beta^{(k)}) - \alpha^*\|_\infty \leq \|g\|_\infty \leq \sqrt{C_1\varepsilon^{(k)}}$. As a result, $\varepsilon = 2\lambda(\|f\|_\infty + \|g\|_\infty) \leq 4\lambda\sqrt{C_1\varepsilon^{(k)}}$.

