# OpenReview forum: "Efficient, Stable, and Analytic Differentiation of the Sinkhorn Loss"
_ICLR.cc/2023/Conference — Submitted to ICLR 2023_

### Official Review · Reviewer_eynF · 2022-10-24

**Confidence:** 4
**Correctness:** 3
**Technical Novelty And Significance:** 2
**Empirical Novelty And Significance:** 2
**Recommendation:** 3

**Clarity, Quality, Novelty And Reproducibility:**

The paper is well-written and easy to follow, and the authors provided code to reproduce their results. However, I have concerns about the quality and novelty of the work as detailed in my major comments in the "Strength and Weakness" section.

**Strength And Weaknesses:**

**Strength:**
1. The problem studied is a highly relevant problem.
1. They provided a convergence analysis of the L-BFGS method on the EOT problem.
1. The advocated L-BFGS + analytic gradient is computationally more efficient than other methods with which they compare.
1. The paper is well-written and easy to follow.

**Major comments:**
1. The novelty of the paper is not super clear to me.
    1. As mentioned by the authors, using L-BFGS to solve the EOT problem is a known practice. One selling point of this paper is the convergence and stability analysis of this algorithm (Theorems 2 - 4). However, this seems to be a practice problem given that L-BFGS has been extensively studied in other (potentially broader) settings. Is the proof technique new? Moreover, why can you conclude $T^{(k)}$ does not underflow from (e)? Don't you need to prove a lower bound for $min T^{(k)}$?
    1. Cuturi et al. (2019) and Cuturi et al. (2020) consider gradients w.r.t data points. Since the cost matrix is constructed from $c(X, g_\theta(Z))$ in the generative modeling application, knowing the gradients w.r.t data points is enough to get the gradients w.r.t $\theta$. Can you also compare your approach with theirs?
    1. You missed an important reference on this topic [1].
1. The numerical experiments are not convincing. In particular,
    1. Fig. 2 does not provide a fair comparison. As mentioned by the authors themselves, one iteration of the L-BFGS algorithm is more time-consuming than one iteration of the Sinkhorn algorithm. Why not use a similar plot as in Fig. 3?
    1. In Table 2, what are the stopping criteria for each of the methods?
    1. In Sections 7.2 and 7.3, while the goal is to compute the Sinkhorn loss, the performance is measured in Wasserstein distance. This mismatch makes it difficult to evaluate the improvement. I would use the Sinkhorn divergence (Feydy et al. '19) instead since this value should decrease as well.
1. I would like to see the full algorithm of the advocated forward and backward passes with complexity analysis.

[1] Tianyi Lin, Nhat Ho, Michael I. Jordan. On the Efficiency of Entropic Regularized Algorithms for Optimal Transport. JMLR (2022).

**Minor comments**
1. The notation $\mu^{-}$ is a bit confusing. I would use $\mu^{-1}$ instead.
1. It would be better to define $s_v$ directly rather than $\tilde s_v$.
1. For matrices $M$ and $T^\star$, their tilde versions are not defined.
1. It would be interesting to know how the marginal constraints of $T^{(k)}$ change with the number of iterations in Theorem 3.
1. In Theorem 4, how large is $\sigma$ typically?

**Summary Of The Paper:**

This paper studies the problem of entropy-regularized optimal transport (EOT). The authors advocate the usage of L-BFGS algorithm to solve the EOT problem instead of the popular Sinkhorn algorithm. They analyze the convergence of L-BFGS on EOT and demonstrate its better stability compared to the Sinkhorn algorithm. They also derive a closed form of the gradient of the Sinkhorn loss w.r.t the cost matrix, leading to efficient computation of the backward pass. Finally, we illustrate their results with numerical examples.

**Summary Of The Review:**

The paper studies a relevant problem and provides some new insights. However, I have some concerns about the quality and novelty of the work as detailed in my major comments in the "Strength and Weakness" section. I thus recommend rejection.

---

> ### Author Response · Authors · 2022-11-15
> **Response to Reviewer eynF**
>
> We thank the reviewer for raising various questions that helped us improve the manuscript. Below are our point-by-point responses.
>
> > The novelty of the paper is not super clear to me.
>
> We have clarified our novelty in a new thread "summary of revision".
>
> > Is the proof technique new?
>
> One major update to the original article is that we have greatly improved the theoretical analysis by providing computable constants for Theorems 2 and 3. This is a novel contribution as it is not a direct consequence of existing L-BFGS theories. Instead, we carefully analyze the dual objective function, and develop new useful bounds for the eigenvalues of its Hessian matrix (Lemma 8), as the constants for Theorem 3 highly rely on these bounds. Such result may be of interest by themselves.
>
> Also, L-BFGS is typically used as a generic optimization method, but we provide richer information specific to OT problems, such as marginal errors, transport plans, etc. Moreover, we provide several results on its numerical stability, which is not common in the existing literature.
>
> > Why can you conclude $T^{(k)}$ does not underflow from (e)? Don't you need to prove a lower bound for $\min T^{(k)}$?
>
> Thanks for raising this question. This is because some entries of the true transport plan $T^*$ can be arbitrarily close to zero, so $T^{(k)}$ also inherits such issues and we cannot provide guarantees on these entries. But on the other hand, we can guarantee that each row/column of $T^*$ must have at least one element that is reasonably large, which is given by $\max_i T_{ij}^*$ and $\max_j T_{ij}^*$. Our results show that these "pivotal entries" for $T^{(k)}$ do not underflow.
>
> > Since the cost matrix is constructed from $c(X,g_\theta(Z))$ in the generative modeling application, knowing the gradients w.r.t data points is enough to get the gradients w.r.t $\theta$.
>
> This is correct, but our derivative $\nabla_M S_\lambda$ is more general and useful. Suppose that there is a hyperparameter in the cost function that is independent of the data. Then to learn this hyperparameter we need more than the gradient w.r.t. the data points.
>
> > Cuturi et al. (2019) and Cuturi et al. (2020) consider gradients w.r.t data points... Can you also compare your approach with theirs?
>
> As far as we know, these methods are used by the OTT-JAX library to compute the implicit differentiation of the Sinkhorn loss, so in our numerical experiment we have already compared with these methods (the "Implicit" method in Section 7 and Appendix B.3, B.4).
>
> > You missed an important reference on this topic [1].
>
> Thanks for the reference. We have cited it in the revision, and have studied the Greenkhorn algorithm in the motivating example (Figure 1) and in one new numerical experiment (Appendix B.2).
>
> > Fig. 2 does not provide a fair comparison.
>
> We shall first clarify that one *gradient evaluation* of L-BFGS is on par with one *iteration* of Sinkhorn-log algorithm, but one L-BFGS *iteration* may consist of multiple gradient evaluations, thus more time-consuming than one Sinkhorn-log iteration. The comparison between Sinkhorn iterations and L-BFGS gradient evaluations is considered fair.
>
> Also in Figure 1, we limit the number of iterations for Sinkhorn-type algorithms, and limit the number of gradient evaluations for L-BFGS.
>
> > In Table 2, what are the stopping criteria for each of the methods?
>
> Thanks for pointing out this issue. We have documented the stopping criteria in the revised Appendix B.3, which is based on the marginal errors. Also we have carefully set the tolerance parameters for different algorithms for fair comparison.
>
> > I would use the Sinkhorn divergence (Feydy et al. '19) instead since this value should decrease as well.
>
> We appreciate this suggestion and have added this metric in the revised Section 7.3 and Appendix B.4.
>
> > I would like to see the full algorithm of the advocated forward and backward passes with complexity analysis.
>
> We have commented in the revision that each gradient evaluation of L-BFGS requires $O(mn)$ operations, and the analytic backward pass costs $O(m^2 n + m^3)$ operations, assuming $n\ge m$.
>
> > Minor comments...
>
> We have changed the notations to reflect these suggestions. We have added the marginal error results in the new Theorem 3, and have added Lemma 8 to provide estimates on the $\sigma$ constant in Theorem 4.

---

### Official Review · Reviewer_7wxm · 2022-10-25

**Confidence:** 4
**Clarity, Quality, Novelty And Reproducibility:** It is well-written paper, easy to fol…
**Correctness:** 4
**Technical Novelty And Significance:** 3
**Empirical Novelty And Significance:** 2
**Recommendation:** 6

**Strength And Weaknesses:**

Strength:

1. The paper presents some useful results (Theorem 1) of computing derivatives of the OT cost between two distribution with respect to the cost matrix. The results could be used in some important settings, i.e generative modeling.

2. The author also presents some theoretical results to support their arguments. More particular, the convergence rate of the objective function, bound of the derivative etc. Those results are presented in Theorem 2,3 and 4.

3. Emperical results show that the method work well and faster than other methods.

Weaknesses:

1. Theorem 2,3 and 4 are helpful but not useful. In particular, index $I$ in Theorem 2 depends on the solution. The rate $r$ in Theorem 3 is unknown, it is supposed to be dependent on $\lambda$.  The same situation in Theorem 4, where $\sigma$, $C_S, C_v$ and $C_u$ are difficult to quantified as a function of the original parameters: the marginal distribution  $a$, $b$ and the cost matrix $M$. That makes it very difficult to compare the order of the complexities of this method and the Sinkhorn algorithm in order to answer the question if this method is faster because of its initialization or it has a larger step to the optimal value etc. Note that there are already some works [1] to show the linear convergence of the objective function of the Sinkhorn algorithm, but it does not improve the upper bound of its complexity [2].

2. Empirical results are limited to some toy examples and small data set (MINIST).

References

 [1] Vladimir Kostic, Saverio Salzo and Massimiliano Pontil. Batch Greenkhorn Algo- rithm for Entropic Regularized Multimarginal Optimal Transport: Linear Rate of Con- vergence and Iteration Complexity. Proceedings of the 39th International Conference on Machine Learning, PMLR 162:11529-11558, 2022.

[2] Dvurechensky, P., Gasnikov, A., and Kroshnin, A. Com- putational optimal trans- port: Complexity by accelerated gradient descent is better than by Sinkhorn’s algorithm. In International conference on machine learning, pp. 1367– 1376, 2018.

**Summary Of The Paper:**

The paper proposes to use L-BFGS algorithm to solve the OT problem by exploiting a loose condition in the Sinkhorn algorithm. They derive a formula to compute the derivative with respect to cost matrix of the transportation cost. Then they prove some results of the convergence of their proposed algorithm.


**Summary Of The Review:**

Results of Theorem 1 is useful in application such as generative modeling, but other results are not.

---

> ### Author Response · Authors · 2022-11-15
> **Response to Reviewer 7wxm**
>
> Thanks for the helpful comments. Below are our point-by-point responses.
>
> > Theorem 2,3 and 4 are helpful but not useful.
>
> One major update of this revision is that we have greatly improved the theoretical analysis by providing computable constants for Theorems 2 and 3. This is a novel contribution as it is not a direct consequence of existing L-BFGS theories. Instead, we carefully analyze the dual objective function, and develop new useful bounds for the eigenvalues of its Hessian matrix (Lemma 8), as the constants for Theorem 3 highly rely on these bounds. The expressions for the constants are given in Appendix A.
>
> For Theorem 4, although at this moment we are not able to give a bound that is free of any unknown quantity, its main merit is that the gradient error is guaranteed to also decay exponentially fast. To the best of our knowledge, such an error anaysis on the backward gradient is new.
>
> > Empirical results are limited to some toy examples and small data set (MINIST).
>
> We remark that the proposed algorithm is mainly used to accelerate the computation and differentiation of the Sinkhorn loss function, which can be viewed as the last layer of a deep generative model. Therefore, the complexity of the problem is mostly determined by the cost matrix, whose dimensions are further determined by the mini-batch sizes. In other words, the computational efficiency of the Sinkhorn loss is neutral to the model architecture prior to the loss function layer. We design the experiments by simulating cost matrices of different sizes and distributions, and the results are generalizable to larger data sets and more complicated model achitectures.
>
> As we have mentioned in the summary of revision, we have added new experiments to justify the efficiency of the proposed algorithms, including the much larger CelebA data set.

---

### Official Review · Reviewer_toCu · 2022-10-25

**Confidence:** 4
**Correctness:** 3
**Technical Novelty And Significance:** 3
**Empirical Novelty And Significance:** 3
**Recommendation:** 6

**Clarity, Quality, Novelty And Reproducibility:**

## Clarity

The paper is overall clear. Theoretical statement are precise and numerical experiments are well described overall.

## Quality

I think this is a solid work overall. Few points remain to be discussed (see Weaknesses and Questions above).

## Novelty

Though tackling an existing problem, the proposed analysis is new to the best of my knowledge. My main concern about novelty would rather be that the work may not be placed in the most "state-of-the-art" setting of computational OT.

## Reproducibility

From a theoretical perspective, the claims are supported by detailed proofs in the appendix. While I could not proofread all of them, I checked the proof of Theorem 1 and did not identify major flaws. The following proofs looked convincing at first glance, though I obviously encourage the authors to carefully proofread the paper as the proofs are quite technical.

On the experimental side, code to reproduce experiments has been anonymously provided and though not tested, I am fairly confident that one could reproduce the numerical results of the paper. Also, the fact that implementation is available in both PyTorch and Jax is a nice bonus.

**Strength And Weaknesses:**

## Strengths

This is a solid work overall. I particularly appreciate that algorithmic aspects (convergence, stability) are supported by precise theoretical results (and not just empirical ones / informal intuition).
I also think that the main idea conveyed by the paper may be impactful when it comes to use OT in ML tasks.

## Weaknesses

- My main concern is that, in some sense, focusing on the loss $\braket{T^\star, M}$ feels somewhat ``outdated''. While it is the seminal loss proposed by Cuturi in 2013, many variants/improvements have been considered since. In particular, I would think that the so-called _Sinkhorn divergence_ [1,2,3] have taken the lead when it comes to introduce OT-based losses in ML. While I guess that a vast majority of this work holds/adapts to that setting, I think it would have been arguably better to focus on it. If there is a precise reason to focus on $\braket{T^\star, M}$ rather than the Sinkhorn divergence, this should be discussed in the paper.
- One may argue that, although fairly convincing, the experimental section is not ``groundbreaking''

## Complementary comments/suggestions/questions

- I think that few statements in the introduction should be made more precise. For instance, "the computational cost of OT quickly becomes formidable" --> give the precise complexity; similarly, it would be nice to recall what is the best theoretical complexity known for the Sinkhorn algorithm.
- It is slightly inaccurate to talk about "**The** optimal solution to (1)" as it may not be unique (in contrast to those of (2), as highlighted after in the paper).
- [minor suggestion] Of course, that is a matter of taste, but I think that using $\eta = \lambda^{-1}$ would alleviate notations and has become more standard in computational OT literature. That would disambiguate claims like "Sinkhorn-type algorithms may be slow to converge, especially for small regularization parameter" (here, one may think that "small regularization parameter" refers to "small $\lambda$", while if my understanding is correct, this should be understood as "small $\lambda^{-1}$").
- Formally speaking, I think that (5) is a _concave maximization problem_ (but of course this is equivalent to a convex minimization problem).
- In Theorem 1, $s_v$ has not been defined if I am correct.
- "exponentiation operation are more expensive than matrix multiplications", is that correct? I would expect term-wise exponentiation to be faster as it is $O(n^2)$ vs $O(n^3)$ overall (unless "matrix multiplications" refers to "matrix-vector multiplications", but even there I would expect the complexity to be at least similar). I also think that both operations can be efficiently performed on a GPU.
- [question] In Theorem 1 and 4, what guarantee that $D$ is invertible (and that $\lambda_{\min}(D) \neq 0$)? I tried to quickly check the appendix but did not find any discussion on that topic (I may have missed it though).
- [suggestion/question] In Theorem 4, it may be more "reassuring" to state/recall that the condition $\epsilon^{k_0} < ... $ is always met since $\epsilon^{(k)} \to 0$ (and is decreasing); assuming the rhs term is not $0$. Related to the above question, do we have any control on how long it takes for this condition to be satisfied?
- In numerical experiments, it is observed that when the regularization parameter $\lambda^{-1}$ is too large, $S$ is a poor approximation of $W$ in sense that it does not go to zero in that setting. In my opinion, this is an additional argument in favor of using the Sinkhorn divergence $\mathrm{SkDiv}$: while for large regularization parameters these divergences will naturally differ from exact OT in general, one still has $\mathrm{SkDiv}(\mu,\nu) = 0 \Leftrightarrow \mu = \nu$ (and more precisely, they metricize the same topology as $W$ for any $\lambda$), so they may be more suited to study the expressiveness/convergence of generative models.
- [typo] In the appendix, both section B.4 and B.5 are called "proof of Theorem 3".

## References

- [1] _On wasserstein two-sample testing and related families of nonparametric tests_, Ramdas et al.
- [2] _Learning generative models with sinkhorn divergences_, Genevay et al.
- [3] _Interpolating between optimal transport and mmd using sinkhorn divergences_, Feydy et al.

**Summary Of The Paper:**

This paper provides an analytic overview of the ``sharp Sinkhorn loss'' and its derivatives with respect both the input weights and the cost matrix (that typically depends on the input locations). Formally, given two discretes probability measures $\mu = \sum_i a_i \delta_{x_i}$ and $\nu = \sum_j b_j \delta_{y_j}$, and a cost $M = (|x_i - y_j|^2)_{ij}$ (or actually any other expression), the considered loss reads

$$ S(M,a,b) = \braket{T^\star, M} $$

where

$$T^\star \in \mathrm{argmin} \braket{T, M} - \eta h(T), $$

where $\eta > 0$ is a regularization parameter, $h$ denotes the entropy function, and the minimization is performed over the transportation polytope between $\mu$ and $\nu$ (that, in this discrete setting, only depends on $a = (a_1,\dots, a_n)$ and $b = (b_1,\dots, b_n)$). A well-known fact is that $T^\star$ can be computed (or at least, estimated) by running the Sinkhorn algorithm, an iterative fixed-point procedure.

In applications (especially in generative models), it is often needed to compute gradients of $S$ with respect to $a,b$ and/or $M$. However, the latter is typically obtained via ``unrolling'' the iterations of Sinkhorn (via automatic differentiation). In contrast, this work computes $T^\star$ relying on a L-BFGS approach instead of the usual iterative loop.

While using L-BFGS to get $T^\star$ is not entirely new, this paper has the following main contributions :
- They study precisely the analytical behavior of the L-BFGS algorithm. The main takeaway are that convergence rates are (asymptotically) the same (i.e. geometric convergence), but the L-BFGS approach can be provably stable. This is an advantage over the standard Sinkhorn loop, and while unstability of the Sinkhorn loop is typically handled by performing iteration in log-domain, this comes at the price of computational efficiency. Here, the proposed approach gets the best of both world, being stable while remaining efficient.
- An analytic expression of $\nabla_M S(M, a,b)$ has been derived, allowing "direct" differentiation rather than unrolling the iterations when---for instance---training a generative model.
- Showcase their approach through some numerical experiments.

**Summary Of The Review:**

I think this is a solid work overall, with potential impact for computational OT practitioners. Nonetheless, few details remain to be clarified/discussed in my opinion before fully supporting it.

---

> ### Author Response · Authors · 2022-11-15
> **Response to Reviewer toCu**
>
> We appreciate the various helpful comments. Below are our point-by-point responses.
>
> > My main concern is that, in some sense, focusing on the loss $\langle T^*,M\rangle$ feels somewhat "outdated"... If there is a precise reason to focus on rather than the Sinkhorn divergence, this should be discussed in the paper.
>
> This is an interesting question and we have the following findings. First, the Sinkhorn divergence defined in [1, 2] is built on top of the Sinkhorn loss in this article. Let $\tilde{S}_\lambda(\mu,\nu)$ and $S_\lambda(\mu,\nu)$ denote the *regularized* and *sharp* Sinkhorn loss, respectively (our definition in Section 2). Then [1] defines the *Sinkhorn divergence* as
>
> $$\tilde{S}_\lambda(\mu,\nu)-0.5\cdot\tilde{S}_\lambda(\mu,\mu)-0.5\cdot\tilde{S}_\lambda(\nu,\nu),$$
>
> and [2] uses the definition
>
> $$2 S_\lambda(\mu,\nu)-S_\lambda(\mu,\mu)-S_\lambda(\nu,\nu).$$
>
> Therefore, the computation and differentiation of the Sinkhorn divergence would finally reduce to those of the Sinkhorn loss $\tilde{S}_\lambda(\mu,\nu)$ or $S_\lambda(\mu,\nu)$, and the algorithms developed in this article can be directly applied to models based on the Sinkhorn divergence.
>
> As for the choice between $\tilde{S}_\lambda(\mu,\nu)$ and $S_\lambda(\mu,\nu)$, it is generally considered that $S_\lambda(\mu,\nu)$ is a better approximation to the Wasserstein distance (see our revised Proposition 1). Therefore, we focus on the $S_\lambda(\mu,\nu)$ version in this article.
>
> Finally, in the revised Section 7.3 and Appendix B.4 we have indeed added Sinkhorn divergence as an alternative metric to evaluate model performance. Also in the revised Appendix B.6, we show experiments that combine Sinkhorn divergence with Wasserstein auto-encoders to train generative models.
>
> [1] Interpolating between optimal transport and mmd using sinkhorn divergences, Feydy et al.
>
> [2] Learning generative models with sinkhorn divergences, Genevay et al.
>
> > One may argue that, although fairly convincing, the experimental section is not "groundbreaking".
>
> We remark that this article focuses on the computational aspect of the Sinkhorn loss, so the performance on the model aspect is standard. Instead, it is more interesting to look at Table 2 and Table 3 that compare the computational time of different methods. In the revised article, we have added convergence tests of algorithms, and it is a bit surprising to us that Sinkhorn's algorithm can be slow to converge with the given accuracy level. This suggests that for serious scientific computing problems that require a relatively high precision, the advocated L-BFGS method is potentially a better candidate.
>
> Also, there is a large difference on the computing time for the backward stage, as is reflected in Table 2 and Table 3.
>
> > Complementary comments/suggestions/questions
>
> The responses below follow the order of the original comments.
>
> - Thanks for the suggestion. We have added such complexity in the revised introduction section.
> - We have corrected such misuses here and elsewhere.
> - We have noted this definition at the beginning of Section 3 and changed the notation globally.
> - We have changed it to " (5) is equivalent to an unconstrained convex optimization problem".
> - We have clarified its definition in the revision.
> - Indeed the comparison is between an $O(mn)$ exponentiation operation and an $O(mn)$ matrix-vector multiplication. We also agree that this difference is minor on modern hardware such as GPUs, which justifies the use of stable algorithms such as Sinkhorn-log and L-BFGS.
> - We have improved the theorem by explictly claiming that $D$ is invertible. Even stronger, we have proved that $D$ is strictly positive definite, with computable bounds on its eigenvalues (Lemma 8).
> - Thanks for the suggestion. We have added this statement in the revision. Theoretically it takes logarithmic time for this condition to hold, based on the exponential rate of $\varepsilon^{(k)}$.
> - We agree that the Sinkhorn divergence would be useful in such cases, and in the revised Section 7.3 and Appendix B.4 we have indeed added Sinkhorn divergence as an alternative metric to evaluate model performance. Since the focus of this article is on computation, and the Sinkhorn divergence essentially is built on top of the Sinkhorn loss, we expect that the divergence version can also benefit from the techniques developed in this article.
> - Thanks for pointing out. We have corrected this typo.

---

### Official Review · Reviewer_sDPq · 2022-10-25

**Confidence:** 4
**Correctness:** 3
**Technical Novelty And Significance:** 2
**Empirical Novelty And Significance:** 2
**Recommendation:** 3

**Clarity, Quality, Novelty And Reproducibility:**

- I thank the authors for joining the code of the numerical experiments. Reproducibility is guaranteed.
- The paper is clear and easy to follow.

**Strength And Weaknesses:**

**Strength**
- LBFGS algorithm to optimize the dual objective function of the Sinkhorn divergence.
- An analytic form of the gradient of the sharp Sinkhorn with respect to the cost function.

**Weakness**
- The authors argue that the transport plan resulting from the L-BFGS algorithm is stable compared to ones from vanilla /stabilized Sinkhorn iterations. For me, this comparison is not fair, since it is only visually; it is mandatory to establish a theoretical result or an extensive empirical study showing this gain of stability. I mean one can show that the Frobenius norm (or any matrix norm) between the LBFGS plan and the unregularized plan is less than the vanilla Sinkhorn plan and the unregularized plan. Alternatively, one can investigate the difference between the objective dual function.
- The motivation behind calculating the gradient cost should be highlighted.
- The numerical experiments on real data are limited.

**Summary Of The Paper:**

The paper focuses on the so-called 'sharp Sinkhorn distance', which can be read as between the Wasserstein distance (without regularization) and the vanilla Sinkhorn divergence. The main contributions are two-fold: (i) the authors propose an L-BFGS algorithm to optimize the dual objective function of the Sinkhorn divergence leading to a stable transport plan (ii) an analytic form of the gradient of the shape Sinkhorn with respect to the input matrix is given. Some numerical experiments to corroborate the theoretical finds are illustrated on synthetic and real datasets.

**Summary Of The Review:**

Overall, the paper lacks some theoretical results or an extensive numerical study to prove the stability of the proposed LBFGS algorithm.

**Superlative phrases**

In the core of the paper, I noticed that there are lots of superlative phrases. I am wondering if this fact is scientifically acceptable.
- Abstract: "superiror performance"
- Page 8: "enjoys superior computational efficiency"
- Page 8: "huge advantage of a closed-form"
- Page 8: "proposed algorithm is the most efficient one"
- Page 9: "highlights the value of the proposed method"

**Typos**
- Page 2: "The optimal solution to (1)" --> "An optimal solution to (1)". Problem (1) could have many solutions.
- Page 2: "satisfies some suitable conditions" --> Add a reference.
- Page 3: "diagonal matix" --> "diagonal matrix".
- Page 5: in Theorem 2: the norm $\||{\gamma^*}\||$ is not specified; is it the $\ell_2$-norm? or $\ell_\infty$-norm? etc.
- Page 6: in Theorem 4: the matrix $D$ is not defined.
- Page 9: "methods achives" --> "methods achieves"
- References: "wasserstein" --> "Wasserstein"
- References: "sinkhorn" --> "Sinkhorn"

---

> ### Author Response · Authors · 2022-11-15
> **Response to Reviewer sDPq**
>
> Thanks for the comments, which greatly helped us improve the manuscript. Our responses to the questions are summarized below.
>
> > This comparison is not fair, since it is only visually; it is mandatory to establish a theoretical result or an extensive empirical study showing this gain of stability.
>
> We appreciate this suggestion and have added experiments in Section B.2. In particular, we simulate data with different distributions, and test various commonly-used algorithms to compute the Sinkhorn loss on the cost matrices. The methods we compare are implemented using the POT library (https://pythonot.github.io/), which is one of the standard packages for OT problems.
>
> The results show that some existing algorithms simply produce NaN values due to numerical instability, or give results with large errors. The advocated L-BFGS solver does not suffer from such issues and results in smaller errors.
>
> On the theoretical aspect, to the best of our knowledge, detailed stability analyses such as explicit bounds on iterates and transport plans are rare in the literature on Sinkhorn's algorithm. Our theoretical results for L-BFGS are new to this field.
>
> > The motivation behind calculating the gradient cost should be highlighted.
>
> If our understanding is correct, the question here is about the motivation behind computing $\nabla_M S_\lambda(M, a, b)$. Please correct us if we misunderstand the question.
>
> We have briefly explained the reason in Section 5. Suppose that we want to train a generative model $g_\theta (Z)$ to match the observed data $X_1,\ldots,X_n$, where $g_\theta(\cdot)$ is a neural network and $Z$ follows a known distribution. Then we can generate data points $Y_j=g_\theta(Z_j)$ and construct a cost matrix $M$ between $X_i$ and $Y_j$. Obviously $M=M_\theta$ depends on $\theta$, so if we want to differentiate the loss function $S_\lambda$ with respect to $\theta$, we need to know $\nabla_M S_\lambda$.
>
> > The numerical experiments on real data are limited.
>
> We remark that the proposed algorithm is mainly used to accelerate the computation and differentiation of the Sinkhorn loss function, which can be viewed as the last layer of a deep generative model. Therefore, the complexity of the problem is mostly determined by the cost matrix, whose dimensions are further determined by the mini-batch sizes. In other words, the computational efficiency of the Sinkhorn loss is neutral to the model architecture prior to the loss function layer. We design the experiments by simulating cost matrices of different sizes and distributions, and the results are generalizable to larger data sets and more complicated model achitectures.
>
> As we have mentioned in the summary of revision, we have added new experiments to justify the efficiency of the proposed algorithms, including the much larger CelebA data set.
>
> > Superlative phrases.
>
> Thanks for raising such issues. We have corrected such phrases and replaced them with more objective statements.
>
> > Typos and other issues.
>
> We have tried our best to correct the typos. For the norm in Theorem 2, we have defined it at the beginning of Section 3 in the revised article. In Theorem 4 we mention that we reuse the symbols in Theorem 1, where the matrix $D$ is defined.

---

### Author Response · Authors · 2022-11-15
**Summary of revision**

## Clarification on novelty

In this article we focus on addressing several computational issues of the sharp Sinkhorn loss. We point out that in existing implementations both the calculation (forward pass) and the differentiation (backward pass) encounter challenges. Then we suggest using L-BFGS to compute the forward loss, and develop an analytic formula for the backward gradient.

What distinguishes this article (with the current revision) from the existing literature is the following:

1. For the forward pass, most existing articles focus on Sinkhorn-type algorithms, but we empirically show that they typically suffer from either stability issues or slow convergence problems. Instead, we rigorously analyze the L-BFGS algorithm on Sinkhorn problems, and prove its convergence and stability.

2. One major novelty in analyzing L-BFGS is that we are able to provide nonasymptotic error bounds with explicit and computable constants. This is achieved by carefully studying the Sinkhorn dual objective function and the eigenvalue structure of its Hessian matrix.

3. L-BFGS is typically used as a generic optimization method, but we provide richer information specific to OT problems, such as marginal errors, transport plans, etc. Moreover, we provide several results on its numerical stability, which is not common in the existing literature.

4. For the backward pass, we have derived an analytic expression for the gradient of the Sinkhorn loss. Numerical experiments show that this closed-form result has visible advantages over existing implementations on computational efficiency.

## Theory

One major update to the original article is that we have greatly improved the theoretical analysis by providing computable constants for Theorems 2 and 3. This is a novel contribution as it is not a direct consequence of existing L-BFGS theories. Instead, we carefully analyze the dual objective function, and develop new useful bounds for the eigenvalues of its Hessian matrix (Lemma 8), as the constants for Theorem 3 highly rely on these bounds. Such result may be of interest by themselves.

## Experiments

In this revision we have expanded the numerical experiments to better validate the proposed algorithms. Some newly added experiments are:

1. Section B.2, which studies the stability and accuracy of several commonly-used forward pass algorithms. This extends the motivating example in Section 3.1, and use formal metrics to evaluate different methods.

2. Section B.3. We clarify the stopping criterion for fair comparison, and also record whether each method actually meets the accuracy threshold. We find that in Table 2 the computing time for competing methods is in fact underestimated, as they do not reach the predefined accuracy. We have added Table 3 that increases the maximum iteration number, and we have observed increased advantage of the proposed method.

3. Section B.4. We have added more experiments on simulated data, and have added the Sinkhorn divergence as an alternative metric to evaluate model performance over time.

We remark that the proposed algorithm is mainly used to accelerate the computation and differentiation of the Sinkhorn loss function, which can be viewed as the last layer of a deep generative model. Therefore, the complexity of the problem is mostly determined by the cost matrix, whose dimensions are further determined by the mini-batch sizes. In other words, the computational efficiency of the Sinkhorn loss is neutral to the model architecture prior to the loss function layer. We design the experiments by simulating cost matrices of different sizes and distributions, and the results are generalizable to larger data sets and more complicated model achitectures.

---

> ### Author Response · Authors · 2022-11-19
> **Update 2: New experiment on CelebA data**
>
> In the second major update we have added a new experiment on the much larger CelebA data [1]. We train a generative model based on the Wasserstein auto-encoder (WAE, [2]), and use the Sinkhorn divergence [3] to serve as the $\mathcal{D}(\cdot,\cdot)$ function in WAE that quantifies the difference of two distributions. Again, our focus is on the computational efficiency of different algorithms. The details of the new experiment is in the revised Appendix B.6.
>
>
> [1] Liu, Ziwei, et al. "Deep learning face attributes in the wild." Proceedings of the IEEE international conference on computer vision, 2015.
>
> [2] Tolstikhin, I., et al. "Wasserstein Auto-Encoders." 6th International Conference on Learning Representations, 2018.
>
> [3] Genevay, Aude, Gabriel Peyré, and Marco Cuturi. "Learning generative models with sinkhorn divergences." International Conference on Artificial Intelligence and Statistics, 2018.

---

### Decision · Program_Chairs · 2023-01-20

**Decision:**

Reject

**Justification For Why Not Higher Score:**

Experimental part is wrong. Presentation of results is misleading.

**Justification For Why Not Lower Score:**

NA

**Metareview: Summary, Strengths And Weaknesses:**

The authors propose a closer look at the differentiation of what is has been called the "sharp" Sinkhorn divergence, which is equal to the dot product of the optimal regularized OT matrix with the cost matrix. This is essentially $\langle T^\star_\lambda(M),M\rangle$. The gradient of that quantity involves differentiating through an argmin, which is then naturally solved with what the authors call an "analytical differentiation". The authors then propose to revisit and study in depth L-BFGS to solve the dual OT problem. They benchmark this approach against vanilla log-Sinkhorn.

Reviewers have liked certain aspects of the paper, but have found some parts underwhelming, notably experiments, novelty, or, for instance, questioning the interest of focusing specifically on the "sharp Sinkhorn divergence". They are overall pushing for a reject.

In addition to the several actionable pieces of feedback provided by reviewers, I think that the authors should consider two aspects to improve their draft:

- I think the authors have to be more straightforward and say that the "analytical derivation" of the gradient of the Sharp Sinkhorn *is* just implicit function differentiation (mentioned in the proof, below Eq. 16). In fact, I fail to see any algorithmic difference between what is carried out in existing toolboxes, and/or the references cited, and what the authors propose. In particular, their analytical formula requires to solve, just like IFT, a linear system. It does not really matter whether the authors rewrite things slightly differently, or change notations. Ultimately, this all boils down to differentiating the dual regularized OT problem optimality conditions, something that is extremely standard at this point, and which has been used in several references. It will be difficult to sell that minor variant of IFT as a contribution in an ICLR paper in 2023, and blurs the overall message.

- The benchmark propose to compare sinkhorn and L-BFGS is not convincing. The authors choose data sizes $n,m$ to be 100~512, or even below in some experiments (Table 2,3 Fig 6). At that scale, the hungarian algorithm or the network simplex is probably faster than all baselines. Worse, for such sizes, the simple overhead of running objects/wrappers (or compiling in the case of jax code, if running jit) makes things extremely blurry. Those sizes are also irrelevant for practical use cases. To be convincing, this section should at the very least look at sizes of 5k and well above. This is 10x more than what the authors have spent their efforts on. More generally, one would expect such benchmarks to replicate settings (e.g. convergence tolerance, epsilon value, data distributions) that are used in real applications.